

# Atmospheric Black Carbon in the metropolitan area of La Paz and El Alto, Bolivia: concentration levels and emission sources

Valeria Mardoñez-Balderrama[1,2,a], Griša Močnik[3,4,5], Marco Pandolfi[6], Robin L. Modini[7], Fernando Velarde[2], Laura Renzi[7,8], Angela Marinoni[8], Jean-Luc Jaffrezo[1], Isabel Moreno R.[2], Diego Aliaga[9], Federico Bianchi[9], Claudia Mohr[7,10,13], Martin Gysel-Beer[7], Patrick Ginot[1], Radovan Krejci[10], Alfred Widensohler[11], Gaëlle Uzu[1], Marcos Andrade[2,12], Paolo Laj[1,9]

[1]Institut des Géosciences de l'Environnement, Université Grenoble Alpes, CNRS, INRE, IRD, Grenoble INP, 38000 Grenoble, France.

[2] Laboratorio de Física de la Atmósfera, Instituto de Investigaciones Físicas, Universidad Mayor de San Andrés, La Paz, Bolivia.

[3] Center for Atmospheric Research, University of Nova Gorica, 5270 Ajdovščina, Slovenia

[4] Haze Instruments d.o.o., 1000 Ljubljana, Slovenia

[5] Jozef Stefan Institute, 1000 Ljubljana, Slovenia

[6] Institute of Environmental Assessment and Water Research (IDAEA-CSIC), Barcelona, 08034, Spain

[7] Laboratory of Atmospheric Chemistry, Paul Scherrer Institute, 5232 Villigen PSI, Switzerland

[8] Institute of Atmospheric Sciences and Climate, National Research Council of Italy (CNR-ISAC), 40129 Bologna, Italy

[9] Institute for Atmospheric and Earth System Research (INAR), and Department of Physics, University of Helsinki, 00014 Helsinki, Finland

[10] Department of Environmental Science & Bolin Centre for Climate Research, Stockholm University, 10691 Stockholm, Sweden

[11]Leibniz Institute for Tropospheric Research (TROPOS), 04318 Leipzig, Germany

[12] Department of Atmospheric and Oceanic Sciences, University of Maryland, College Park, MD 20742, United States of America

[13] Department of Environmental Systems Science, ETH Zurich, 8092 Zurich, Switzerland

[a] now at: Institute of Atmospheric Sciences and Climate, National Research Council of Italy (CNR-ISAC), 40129 Bologna, Italy

*Correspondence to*: Valeria Mardoñez-Balderrama (v.mardonez@isac.cnr.it)

**Abstract.** Black carbon (BC) is a major component of sub-micron particulate matter (PM) with significant health and climate impacts. Many cities in emerging countries lack comprehensive knowledge about BC emissions and exposure levels. This study investigates BC concentration levels, identify its emission sources, and characterize the optical properties of BC at urban background sites of the two largest high-altitude Bolivian cities: La Paz (LP) (3600 m above sea level) and El Alto (EA) (4050 m a.s.l.) where atmospheric oxygen levels and intense radiation may affect BC production. The study relies on concurrent measurements of equivalent black carbon (eBC), elemental carbon (EC), and refractory black carbon (rBC), and their comparison with analogous data collected at the nearby Global Atmosphere Watch-Chacaltaya station (5240 m a.s.l). The performance of two independent source-apportionment techniques was compared: a bilinear model and a least squares multilinear regression (MLR). Maximum eBC concentrations were observed during the local dry season (LP: eBC=1.5±1.6



μg m$^{-3}$; EA: 1.9±2.0 μg m$^{-3}$). While eBC concentrations are lower at the mountain station, daily transport from urban areas is

evident. Average mass absorption cross sections of 6.6-8.2 m$^2$g$^{-1}$ were found in the urban area at 637 nm. Both source apportionment methods exhibited a reasonable level of agreement in the contribution of biomass burning (BB) to absorption. The MLR method allowed the estimation of the contribution and the source-specific optical properties for multiple sources including open waste burning.

## 1 Introduction

Black carbon (BC), a particulate byproduct of hydrocarbon-based fuel combustion, is not only considered one of the most important pollutants contributing to climate change (Bond et al., 2013), but also a threat to air quality and therefore to human health. Named as BC due to its dark appearance, it possesses the ability of absorbing radiation over a wide spectral range. This makes BC a very important short-lived climate forcing agent (Bond et al., 2013). Typically introduced into the atmosphere by vehicular, industrial and biomass burning emissions (whether for residential heating, agricultural purposes, or in wildfires),

BC is often used as a good tracer for continental emissions when monitoring air quality (Seinfeld and Pandis, 2016; Bockhorn, 2013; Subramanian et al., 2006).

Due to their size, fine and ultrafine particles containing BC can be inhaled deeply into the lungs, reaching the blood stream and generating reactive oxygen species (ROS) (Janssen, et al., 2012; Janssen et al., 2014). This in turn can decrease lung function (Suglia et al., 2008), worsen preexisting cardiovascular conditions (Nichols et al., 2013) and increase the risk of

chronic obstructive pulmonary disease hospitalizations and mortality (Gan et al., 2013). Moreover, due to its agglomerate structure and large surface area to volume ratio, BC is an efficient carrier of carcinogenic and mutagenic organic species (Moosmüller et al., 2009; NTP, 2011). In consequence, BC has become a pollutant of big concern in terms of air quality.

The inhalation of BC has been associated with different respiratory and cardiovascular diseases such as asthma, lung cancer and cardiac arrest (US EPA, 2011), and high altitude can increase the risk of exposure to atmospheric pollutants. People in the

Andean highlands have developed different physiological mechanisms to cope with hypoxia. Amongst them are an increased lung capacity and an increased hemoglobin concentration in their blood. Even though, their resting ventilation has shown to remain the same as for people living at sea level, conditions that require a higher ventilation rate (e.g. physical activities, pregnancy) put highlanders at greater risk (Beall, 2007; Exposure Factors Handbook, 2022; Julian and Moore, 2019).

The formation of BC particles during combustion depends not only on the amount and type of fuel burnt but also on the amount

of oxygen available in the bonding process of carbon and oxygen atoms when forming CO and $CO_2$ molecules. Hence, the reduced oxygen availability for combustion in high-altitude cities is expected to increase the production rate of carbonaceous particles (Seinfeld and Pandis, 2016). Furthermore, simulated and experimental studies have shown that the amount of PM particles produced by a diesel-fueled engine can increase from 1.2 to 4 times for an altitude up to 1800 m above sea level compared to the production of particles at sea level (He et al., 2011; Yu et al., 2014; Human et al., 1990; Chaffin and Ullman,

1994; Graboski and McCormick, 1996).



Moreover, fast growing cities are often subject to a degrading air quality resulting from a growing vehicular fleet, especially in low and middle-income countries. Air quality studies in Latin-American high-altitude cities have shown that Elemental Carbon (EC, closely related to BC) and Organic Aerosol particles (OA) account for about 60% of the $PM_{10}$ mass concentrations in Bogota (2550 -2620 m a.s.l.) (Ramírez et al., 2018), with average EC concentrations of $3.25 \pm 1.59$ μg m$^{-3}$. In Quito (2850

m a.s.l.), traffic emissions have been shown to be responsible for almost 46% of the annual PM emissions (Raysoni et al., 2017). In the case of Mexico City (2280 m a.s.l.), Peralta et al. (2019) reported annual BC levels of 2.95 μg m$^{-3}$, contributing to 16-20% of total $PM_{2.5}$ concentrations.

La Paz (between 3200-3600 m a.s.l.) and El Alto (4050 m a.s.l.) are two high-altitude Bolivian cities located in the Bolivian Andean region that form the second largest metropolitan area in the country. High altitude, and hence lower oxygen

concentrations, can make the combustion processes even more complex. As recently shown by Mardoñez et al. (2023) and Wiedensohler et al. (2018), the air quality in these cities is predominantly influenced by local emissions, with vehicular traffic (>80% powered by gasoline) as the main source of absorbing aerosol particles, and responsible for 20-30% of the measured $PM_{10}$ concentrations in both cities.  However, the cities are also subject to regional sources of pollution such as agricultural biomass burning, which is transported from the valleys and low lands across the Andes (Chauvigne et al., 2019; Mardoñez et

al., 2023), and represents an important source of pollution at a regional level (Mataveli et al., 2021). In contrast, residential biomass burning for heating purposes is not a common practice in the region. Moreover, in Mardonez et al. (2023) was reported the presence of a third source of combustion related to open waste burning that contributes to the measured EC mass concentrations in the area and could potentially have an influence in the locally measured absorption.

Improving air quality in La Paz and El Alto is not a trivial task, and the effectiveness of any air quality action plan must be

supported by sound scientific understanding of the lifecycle of the main pollutants in a high altitude and high solar radiation environment. Knowledge of BC emission sources and of its characteristics in the urban environment is key to assess the impact of air quality on health for the specific case of these high-altitude cities. Moreover, the conurbation of La Paz-El Alto (LP-EA) is located less than 20 km away from one of the three Global Atmospheric Watch (GAW) global stations in South America, the Chacaltaya mountain station (CHC-GAW). As previously evidenced by Wiedensohler et al. (2018), this station is often

under the direct influence of the urban emissions for at least a few hours throughout the day, as the urban planetary boundary layer (PBL) develops in conjunction with advective winds that allow the fast transportation of urban local pollution towards the mountain station. Therefore, the characterization of the urban emissions and their transport to CHC-GAW is a crucial task to be able to distinguish between the influence of the urban emissions and the long-range transport of air masses reaching CHC-GAW.

The aims of this study are to contribute to document the atmospheric concentration, the variability, and the physical properties of BC in the unique LP-EA conurbation as well as at the global CHC-GAW station. It is also intended to determine the contribution of local and regional sources of BC in the urban area. To do so, this work makes use of a two-year record of BC and other pollutants measured at two urban background sites and at the mountain CHC-GAW station. It is also intended to provide a spatial description of the BC concentrations, and to explore the effect of a half-kilometer altitude difference and





different topographical characteristics between La Paz and El Alto, thus paving the way for future studies on the potential health effects of air pollution in both cities.

## 2 Method

### 2.1 Site description

Having started as an extension of La Paz, El Alto has rapidly become the second largest city in Bolivia in terms of population,
hosting over 1.1 million inhabitants (INE, 2012-2022, 2022). It is situated on the plateau formed between the two branches of the Andes, a very high, open, flat, and dry (hence dusty) area, with plenty of space for urban expansion. Hosting the second largest international airport in the country, El Alto constitutes one of the most important connections of the Bolivian seat of government (La Paz) to other regions within and outside the country. In contrast, La Paz is located in a river valley formed between the Altiplano plateau and the Oriental branch of the Andes. Characterized by its hilly topography, it hosts over 950
thousand inhabitants (INE, 2012-2022, 2022), with limited space for urban growth.

Meteorological seasons in LP-EA shift between a dry (May to August, austral winter) and a wet (December to March, austral summer) period, with relatively low temperatures throughout the year. However, the difference in altitude and topography produces very different local meteorological conditions. The mean annual temperature in El Alto is 8°C, whereas in La Paz it is 13°C. During the austral summer, mean temperatures only increase by one-degree with mean temperature amplitudes of 10
and 12 degrees along the day in El Alto and La Paz, respectively. In contrast, during the austral winter mean temperatures decrease by two degrees in both cities, and the diurnal temperature amplitudes increase to 17 and 15 degrees. The average annual rainfall between 2016 and 2018 was 470 and 600 mm/year in El Alto and La Paz, respectively. Although not very frequent, precipitation in the form of snow is possible in El Alto during the transition periods. Moreover, wind patterns in La Paz are constrained by the North-South topographic features of the river valley. Average atmospheric pressure at La Paz and
El Alto are around 664hPa and 630 hPa, respectively.

### 2.2 Sampling sites

Urban background sampling sites were chosen in each city to obtain representative measurements of the base state of the air quality. This was not the first campaign that addressed BC concentrations in La Paz and El Alto. In 2012, a shorter campaign provided the first characterization of BC in this high-altitude conurbation and its transport towards the CHC-GAW
(Wiedensohler et al., 2018). The results of the short campaign provided insight to the present long-term campaign, and the same urban background sampling site in El Alto was maintained. In La Paz, the traffic sampling site described in Wiedensohler et al. (2018) was transferred to a more secluded site for the present study, away from direct vehicular emissions. The distance between the sampling sites LP-EA is approximately 8 km, and they are located 17-19 km from the CHC-GAW global monitoring station (Fig. 1) as described in Bianchi et al. (2022).





The La Paz measurement site (LP) was installed on the rooftop of the four-story building of the city's Museum Pipiripi (Espacio Interactivo Memoria y Futuro Pipiripi: 16.5013°S, 68.1259°W, 3600 m a.s.l.). This municipal building is located on a small hilltop in downtown La Paz and is approximately 70 m and 45 m from the nearest road (horizontal and vertical distance, respectively). Within a 1 km radius around the station, the LP site is surrounded by many busy roads and dense residential areas. However, the immediate surroundings (~100 m radius) are covered by green areas, with a parking lot for municipality

buses at the base of the hill.

The El Alto station (EA) was placed at the ground level of the El Alto International Airport's meteorological observatory (16.5100° S, 68.1987° W, 4025 m a.s.l.). The building of the meteorological station is in an area restricted for the general public, 250 m away from the airport runway. The shortest distance to a major road is 500 m as was described elsewhere (Wiedensohler et al., 2018; Mardoñez et al., 2023). The station is surrounded by approximately 3.5 km$^2$ of an empty, dry, and

arid field.

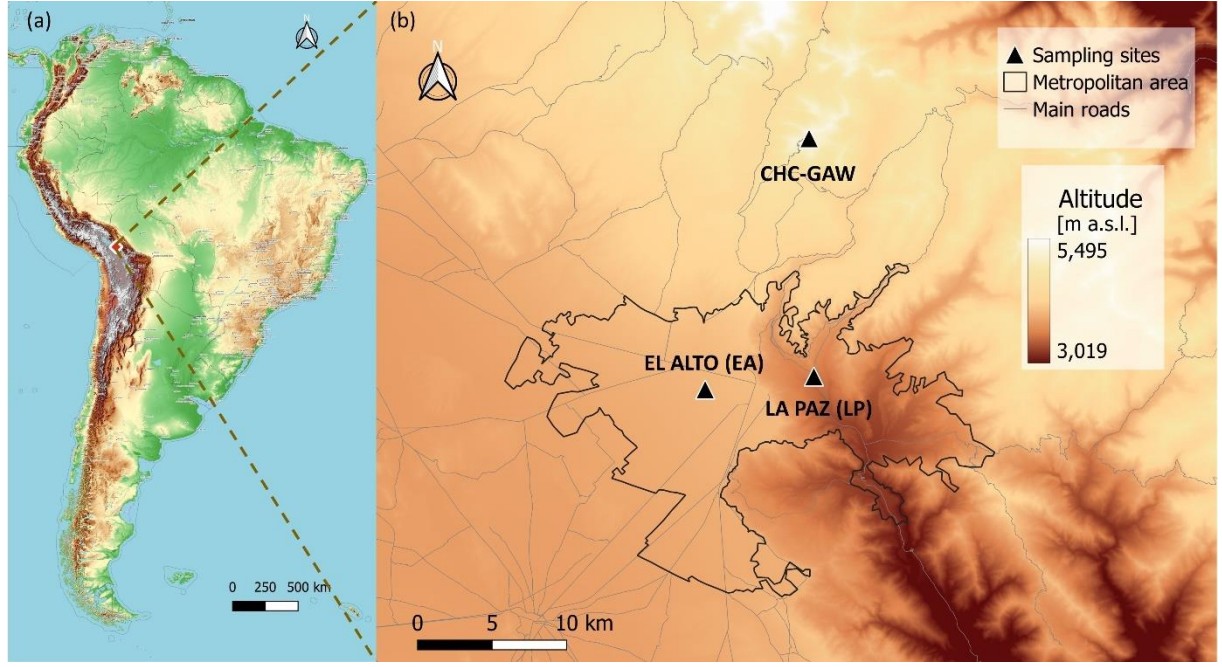

**Figure 1: (a) Geographical location of the sampling sites. (b) The locations of La Paz (LP) and El Alto (EA) have been enlarged and positioned with respect to the regional Chacaltaya GAW monitoring station (CHC-GAW). The color scale represents the altitude above sea level. ©OpenTopoMap (CC-BY-SA) (Mardoñez et al. 2023).**

Both urban stations were equipped with a basic meteorological station with wind speed and direction provided at 15-min time resolution by anemometers placed on the rooftops of the buildings where the rest of the instruments were installed. Other meteorological variables were measured and provided by the Airport's air navigation administration (Administración de Aeropuertos y Servicios Auxiliares a la Navegación Aérea, AASANA) with a 1-hour time resolution, for El Alto, and from the National Meteorology and Hydrology Service (Servicio Nacional de Meteorología e Hidrología, SENAMHI) (SENAMHI,

n.d.) for La Paz at a daily time resolution.




## 2.3 Sampling methods

The measurement campaigns took place between April 2016 and June 2018 with a similar instrumental setup at both urban background sites (LP and EA). Several simultaneous online and offline measurements of chemical and physical aerosol particle properties were continuously performed at both sites during this period, along with measurements of meteorological
parameters. In addition, continuous measurements of atmospheric parameters are taken at the CHC-GAW station since 2012 (Table 1).

**Table 1. Summary table of the instrumentation placed at each of the stations, the analyzed period and time resolution. XR refers to the extended range version of the SP2 (SP2-XR).**

| | *El Alto* | *La Paz* | *CHC-GAW* |
|---|---|---|---|
| *Aethalometer AE33* | Apr 2016 - Sep 2017 (1-min) | X | X |
| *Aethalometer AE31* | Jan 2018 - Jun 2018 (5-min) | Apr 2016 - Jun 2018 (5-min) | Apr 2016 - Jun 2018 (5-min) |
| *$PM_{10}$ filter samples* | Apr 2016 - Jun 2017 (24-h every 3rd day) | Apr 2016 - Jun 2018 (24-h every 3rd day)[1] | Apr 2016 - Aug 2017 (24h or 23:00-08:00 during a week) |
| *$PM_{2.5}$ filter samples* | Jun 2017 - Jun 2018 (24-h every 3rd day) | Jun 2017 - Jun 2018 (24-h every 3rd day) | X |
| *Single particle soot photometer (SP2)* | SP2-XR Apr 2018 - May 2018 (1-hour) | SP2-XR Apr 2018 - May 2018 (1-hour) | SP2-C Apr 2018 - May 2018 (1-hour) |
| *Meteorology* | Apr 2016 - Jun 2018 (1-hour) | Apr 2016 - Jun 2018 (1-day) | Apr 2016 - Jun 2018 (1-hour) |

[1]After June 2017, only 25 $PM_{10}$ filter samples were intermittently collected due to technical problems.

### 2.3.1 High volume Samplers

High-Volume Samplers (MCV CAV-A/mb) were used at both sites to collect aerosol particles on quartz fiber-filters for later analysis. 24-hour filter samples were taken every 3 to 4 days at a flow rate of 30 m$^2$h$^{-1}$ using $PM_{10}$ heads (MCV PM1025UNE) during the first 15 months of sampling at both sites. For the second year, the head was replaced with a $PM_{2.5}$ inlet (MCV PM1025UNE). Additionally, in La Paz, a second high-volume sampler was added during the second year of the campaign to
collect samples of particles with aerodynamic diameters smaller than 10 and 2.5 µm. Sampling always started at 9:00 at both sites. During the campaign, a total of 422 filters were collected between both sites, which were later weighed and analyzed for



over 180 different chemical species, including EC, OC (through thermal-optical analysis (TOA) using a Sunset instrument and the EUSAAR2 protocol) and several organic and inorganic source tracers. A more detailed description of the methodology and protocols can be found in Mardoñez et al. (2023). The ambient concentrations obtained from the collected filters were then
multiplied by a factor of 1.66 and 1.60 in El Alto and La Paz, respectively, to convert them to STP conditions ($\bar{T}$=273 K, $\bar{P}$=1013.5 hPa).

Weekly PM$_{10}$ filter samples were also collected at the summit of Mount Chacaltaya between April 2016 and August 2017, as described by Moreno et al., (2024). The majority of the collected samples (42 out of 50 samples) were nocturnal samples (23:00-8:00/23:00-9:00), outside the hours when the station is directly influenced by the metropolitan area.

### 2.3.2 Single Particle Soot Photometer

The measurement period in the urban area ended with a one-month measurement period where refractory black carbon concentrations (rBC) were measured in LP and EA with Extended Range Single-Particle Soot Photometers (SP2-XR, Droplet Measurement Technologies). The SP2-XR is a new, more compact version of the original SP2 instrument (Baumgardner et al., 2004; Schwarz et al., 2006). During the same period, rBC concentrations were also measured at the CHC-GAW station
with an SP2 (SP2 revision C). The SP2 and SP2-XR provide real-time measurements of the optical size and rBC mass of individual particles, based on measured elastic scattering and laser-induced incandescence signals, respectively. In the present study, all the SP2-based instruments were calibrated with fullerene soot (Baumgardner et al., 2012) size-selected with a differential mobility analyzer (with corresponding masses determined using the effective density approximation of Gysel et al. (2011), and the instrument operation and data analysis procedures were performed following Laborde et al. (2012). The LP
and EA stations were each equipped with an SP2-XR unit and the CHC-GAW station with an SP2-C unit between April and May 2018, from which hourly rBC mass concentrations were retrieved and reported here. A more detailed description of the instruments and the analysis of the measurements can be found in Modini et al (in prep.) and Renzi et al (in prep.).

### 2.3.3 Aethalometers

Aethalometers are filter-based absorption photometers that measure light attenuation by atmospheric aerosol particles at 7
wavelengths in the visible/near-visible spectrum (370, 470, 520, 590, 660, 880, 950 nm). In this study we made use of the measurements taken with these instruments to determine the spectral dependence of light absorption. In addition, the measurements at 880 nm were used to obtain an estimate of the black carbon mass concentration. We followed the recommendation by Petzold et al. (2013) and use the term equivalent black carbon (eBC) to emphasize that this is an operationally defined quantity.
The Aethalometer, model AE33 (Magee Scientific, Drinovec et al., 2015, Tape: Pallflex TFE-coated glass fiber T60A20 /M8020), was used at the EA station at the minimum 1-minute time resolution provided by the instrument (Apr 2016 – Sep 2017). Due to technical problems, the instrument had to be replaced by an Aethalometer AE31 (Magee Scientific, Arnott et al., 2005, Tape: Pallflex Quartz fiber Q250F) for the end of the campaign (Jan – Jun 2018). The same model, AE31, was





operated in La Paz during the entire campaign at its minimum 5-minute time resolution (June 2016- June 2018). $PM_{10}$ inlets

were used at the front of the sampling lines at both sites throughout the campaign, except for the period from April 2017 to September 2017 where whole air was sampled at El Alto station. The $PM_{10}$ heads were installed at 2 m and 6 m above the instrument level for LP and EA, respectively. The instruments were set to report BC concentrations at standard conditions of temperature and pressure (STP) at both urban sites. In addition, the data recorded between April 2016 and July 2018 by an aethalometer AE31 at CHC-GAW station were downloaded from the EBAS database (http://ebas.nilu.no/) and were included

in the analysis.

As for all filter-based instruments, instrumental artifacts needed to be accounted for in the post processing data analysis. Thus, the necessary corrections are then applied to the non-corrected attenuation coefficients ($b_{ATN,NC}$) that are proportional to the change in attenuation (ATN) as defined in Eq. (1)

$$b_{ATN,NC}(\lambda) = \frac{A}{Q \cdot 100} \cdot \frac{ATN_t(\lambda) - ATN_{t-\Delta t}(\lambda)}{\Delta t}, \tag{1}$$

where ATN corresponds to the logarithm of the light intensity ratio reaching the detector after passing through the loaded filter, compared to the attenuation of the light passing through a clean portion of the filter; A is the area of the filter onto which particles are collected ($A_{AE31}$= 0.5 cm$^2$; $A_{AE33}$= 0.785 cm$^2$) and Q is the sample flow rate. These attenuation coefficients can be traced back from the BC mass concentrations reported by both models of aethalometer as shown in Eq. (2) and Eq. (3):

$$b_{ATN,NC,AE31}(\lambda) = BC_{AE31}(\lambda) \cdot \sigma_{AE31}(\lambda) = BC_{AE31}(\lambda) \cdot C_{f0,AE31} \cdot MAC_{AE}(\lambda), \tag{2}$$

$$b_{ATN,NC,AE33}(\lambda) = BC_{AE33}(\lambda) \cdot \sigma_{AE33}(\lambda) \cdot \left(1 - k(\lambda) \cdot ATN(\lambda)\right) = BC_{AE33}(\lambda) \cdot C_{f0,AE33} \cdot MAC_{AE}(\lambda) \cdot \left(1 - k(\lambda) \cdot ATN(\lambda)\right), \tag{3}$$

where $\sigma_{AE31}(\lambda)$ are the mass attenuation coefficients, also expressed as the product of the default correction factor for multiple scattering ($C_{f0}$, which depends on the model of aethalometer, and the filter tape used (in the present study: $C_{f0, AE31}$= 2.14, $C_{f0, AE33}$= 1.57), and the default mass absorption cross-sections established by the manufacturer for each of the seven wavelengths

($MAC_{AE,370nm}$= 18.47 [m$^2$g$^{-1}$]; $MAC_{AE,470nm}$= 14.54 [m$^2$g$^{-1}$]; $MAC_{AE,520nm}$= 13.14 [m$^2$g$^{-1}$]; $MAC_{AE,590nm}$= 11.58 [m$^2$g$^{-1}$]; $MAC_{AE,660nm}$= 10.35 [m$^2$g$^{-1}$]; $MAC_{AE,880nm}$= 7.77 [m$^2$g$^{-1}$]; $MAC_{AE,950nm}$= 7.19 [m$^2$g$^{-1}$]. In Eq. (3), $k$ is the loading correction factor reported by the instrument (Drinovec et al., 2015) which will be described in more detail below.

The measurements taken with the AE33 photometer exhibited notable noise levels when operated at the minimum time resolution (1 min). To address this issue, we applied the noise reduction technique outlined by Backman et al. (2017) and

Springston and Sedlacek (2007), resulting in a significant reduction in noise while preserving the original temporal resolution. Briefly, the latter studies proposed to increase the period Δt from the original sampling time resolution to a higher time interval in Eq. (1). Therefore, $b_{ATN,NC}$ were recalculated using a Δt=30 min for AE33 to efficiently reduce the noise of the data set. This procedure was only applied to $b_{ATN,NC}$ measured within the same filter-spot, hence, the Δt of the first 30 minutes of every new filter-spot were defined as Δt=t-t$_0$.

As the filter gets loaded, particles deposited on the filter may interact with the incoming light beam increasingly reducing the amount of light reaching the detector. This artifact is also known as "loading effect" and needs to be accounted for. The way



this correction is performed constitutes the main difference between models AE31 and AE33. The aethalometer AE33 performs the loading correction online, thanks to its dual spot design (Drinovec et al., 2015). In contrast, measurements taken with the aethalometer AE31 need to be corrected as part of the post processing data analysis (Virkkula et al., 2015). However, an overcompensation of the loading effect was observed in the AE33 data output, therefore it was not used. Instead, the following correction scheme proposed by Virkkula et al. (2015) and Drinovec et al. (2015) was used to correct the data obtained from both aethalometer model (Eq. (4)):

$$b_{ATN}(\lambda) = \frac{b_{ATN,NC}(\lambda)}{\left(1 + \hat{k}(\lambda) \cdot ATN_t(\lambda)\right)}, \tag{4}$$

where $\hat{k}$ are the monthly loading correction factors, defined as the ratio of the slope over the intercept of the linear fit of the non-corrected BC mass concentrations (reported by the instrument) vs ATN (Drinovec et al., 2015).

Multiwavelength absorption coefficients ($b_{abs}(\lambda)$) are typically estimated from $b_{ATN}(\lambda)$ by correcting the latter for multiple the light scattering taking place within the filter fibers or between deposited particles and the filter fibers. This is done by normalizing $b_{ATN}(\lambda)$ by the multiple scattering correction factor ($C_f$), which depends on both the properties of the filter matrix and the optical properties of aerosol population (Eq. (5)). $C_f$ is often considered independent of $\lambda$ although that is likely not the case for AE33 (Yus-Díez et al., 2021).

$$b_{abs}(\lambda) = \frac{b_{ATN}(\lambda)}{C_f}, \tag{5}$$

Several studies have shown that the default $C_{f0}$ factors provided by the manufacturer often underestimates the impact of multiple scattering happening in the filter matrix for both models of aethalometers when measuring ambient aerosol (Yus-Díez et al., 2021; Müller, 2015; Bernardoni et al., 2021; Valentini et al., 2020). Thus, a local $C_f$ =3.08 was estimated for the AE31 operating in the urban stations. This value resulted from a Deming linear regression of the AE31 hourly attenuation coefficients measured at CHC-GAW station and the absorption coefficients measured simultaneously by a Multi Angle Absorption Photometer (MAAP) also operating continuously at the mountain station (Eq. (6), Fig. S1 in the Supplement), forcing the intercept through zero, as determined by previous studies (Valentini et al., 2020; Yus-Díez et al., 2021; Bernardoni et al., 2021):

$$b_{ATN,AE31,CHC}(637\ nm) = C_{f,AE31} \cdot 1.05 \cdot b_{abs,MAAP,CHC}(637\ nm), \tag{6}$$

The factor 1.05 included in Eq. (6) accounts for the difference between the measuring wavelength reported by the MAAP manufacturer (670 nm) and the actual wavelength (637 nm) measured by Müller et al. (2011). The attenuation coefficients measured by the AE31 at CHC-GAW and included in Eq. (6) were interpolated to MAAP's wavelength using the calculated AAE corresponding to each data point (mean $AAE_{CHC-GAW}$ = 1.0±0.5). The extrapolation of this calculated $C_f$ factor to the urban background stations is done under the assumption that the properties of the urban aerosol do not change drastically on their transport to CHC-GAW. Hence, only measurements acquired during the time interval from 10:00 to 16:00 were included in the calculation of $C_f$. This specific time frame corresponds to a period in which the CHC-GAW station is typically under the influence of the urban PBL, as previously demonstrated by Andrade et al. (2015).





A reference instrument was not available to assess the cross-sensitivity to scattering of AE33 at EA, for which we opted for
selecting a suitable literature $C_f$ factor. Systematic higher $b_{abs}$ were observed at EA compared to LP when different models of
aethalometers were employed, whereas a negligible difference was observed when using AE31 aethalometers. Moreover, no
significant differences in the average concentrations of EC were found among the sites. This indicated that the observed
difference was rather associated to differences between models AE33 and AE31. Therefore, a $C_f$=2.78 was set for AE33, for
it brought the $b_{abs}$ estimated from both models of aethalometer closer together. The chosen $C_f$ was reported by Bernardoni et
al. (2021) at an urban background station in Milan during winter 2018 using the same type of filter tape as AE33 in the present
study (T60A20), however, it is 21-40% larger than the values reported by Yus-Díez et al. (2021) for an urban background
station in Barcelona using the same tape. Moreover, the $C_f$ values applied in this study are significantly larger than the default
values given by the manufacturer of 2.14 and 1.57 for the AE31 and AE33, respectively. Therefore, the reported $b_{abs}$ in the
present study are smaller than what they would be using the factory default $C_f$ values. The $b_{abs}$ estimated using the $C_f$ factors
selected for each model of AE31 decrease the instrumental difference to roughly 27%.

## 2.4 Intrinsic properties of BC

### 2.4.1 Absorption Angstrom Exponent (AAE)

The absorption efficiency of particulate matter varies throughout the visible spectrum of radiation and is determined by the
chemical and physical properties of the aerosol particles. This wavelength dependency can often be described in good
approximation by a power law with the Absorption Ångström Exponent (AAE) (Moosmüller et al., 2011) as exponent.
Orthogonal non-linear least squares regressions were used to describe the power-law dependency of absorption with
wavelength, Eq. (7):

$$b_{abs}(\lambda) = a \cdot \lambda^{-AAE}, \tag{7}$$

Only wavelengths 2 to 7 (470 – 950 nm) were used in the regression such that the extreme wavelengths match the pair used
for the BC source apportionment (Sect. 2.5.1). The obtained time series of AAE displayed the same noisy pattern as observed
in the BC time series prior the noise reduction, therefore only AAE corresponding to absorption coefficients $b_{abs,880}$>2 Mm$^{-1}$
were considered in the rest of the analysis for the urban sites. In the case of CHC-GAW, no lower cut was applied to the AAE
values since average levels of $b_{abs}$ were close to the threshold.

Among the major components of PM contributing to the absorption of visible radiation, AAE ~1 are typically observed in BC
particles coming from vehicular emissions (Bond et al., 2013). In contrast, Brown Carbon (BrC, a combustion product
containing organic material) has a stronger absorbance at shorter wavelengths. Hence, a wider range of AAE values have been
observed associated to BrC, 0.9 to 2.2 depending on the type of soil and fuel burned, as well as on the combustion conditions
(Kirchstetter et al., 2004; Sandradewi et al., 2008; Helin et al., 2021). Finally, dust particles have also been found to be capable
of absorbing visible radiation depending on their chemical composition. Dust absorbs visible radiation more efficiently at



shorter wavelengths, therefore AAE values usually > 2 are associated to these particles (Bergstrom et al., 2007; Caponi et al., 2017).

### 2.4.2 Mass Absorption Cross-section (MAC)

The mass absorption cross-section (MAC) is an intrinsic property of absorbing aerosol particles that describes their absorption efficiency per unit of mass. This property depends on wavelength, aerosol type and particle mixing state, and constitutes the

bridge between the optical and physical properties of absorbing aerosol particles. The MAC, thus, provides an insight of the particle mixing state (which is a function of aerosol ageing) and is defined as follows:

$$MAC(\lambda) \ [m^2 g^{-1}] = \frac{b_{abs}(\lambda) \ [Mm^{-1}]}{m \ [\mu g \ m^{-3}]}, \tag{8}$$

where $b_{abs}$ is the absorption coefficient at wavelength $\lambda$, and m is the mass concentration of the absorbing particles. In the present study the MAC coefficients of BC were estimated by using $b_{abs}$ (880 nm) in Eq. (8) (since at 880 nm absorption is

dominated by BC), and the mass concentrations of refractory BC measured by thermal-optical methods (EC) and by laser-induced incandescence (rBC). For the complete extent of the campaign, daily $MAC_{EC}$ were estimated using daily averaged $b_{abs}$ (880 nm) in Eq. (8). Similarly, hourly $MAC_{rBC}$ were calculated for the period in which the high-time resolution rBC was available, using hourly averages of $b_{abs}$ (880 nm). For comparability with other studies, the obtained MAC values were interpolated to other wavelengths typically reported in the literature (550 and 637 nm) by interpolating the absorption

coefficients through their corresponding absorption Ångström exponent.

When independent measurements of BC mass concentrations are not available, Eq. (8) also allows the estimation of the equivalent BC (eBC) mass concentrations, if a representative $MAC_{eBC}$ is defined. Then eBC can be understood as the amount of BC necessary to absorb an equivalent amount of radiation as the one reported by the estimated $b_{abs}$. When measuring absorption coefficients using aethalometers, eBC is typically reported at 880 nm using the default manufacturer

$MAC_{AE,880nm}$=7.77 [$m^2 g^{-1}$]. Then, Eq. (8) becomes:

$$eBC = \frac{b_{abs}(880 \ nm)}{MAC_{AE,880nm}}, \tag{9}$$

### 2.5 Description of methods for performing Black Carbon source apportionment

### 2.5.1 Aethalometer method

Based on the differences in the source-specific AAE, a bilinear regression model was proposed by Sandradewi et al., 2008 in

order to determine the contribution of each source to total absorption. Generally, this technique is used to apportion the contributions of vehicular emissions and biomass burning emissions or dust (Lanz et al., 2008; Sandradewi et al., 2008; Harrison et al., 2012; Zotter et al., 2017). This method is also known in the literature as the "Aethalometer method".

Considering that:

$$b_{abs}(\lambda_i) = b_{abs,TR}(\lambda_i) + b_{abs,BB}(\lambda_i), \tag{10}$$



where the subscripts TR and BB represent absorption due to traffic and biomass burning, respectively, and the power law

dependence of $b_{abs}$ with wavelength described in Eq. (7):

$$\frac{b_{abs,TR}(\lambda_1)}{b_{abs,TR}(\lambda_2)} = \left(\frac{\lambda_1}{\lambda_2}\right)^{-AAE_{TR}}, \tag{11}$$

$$\frac{b_{abs,BB}(\lambda_1)}{b_{abs,BB}(\lambda_2)} = \left(\frac{\lambda_1}{\lambda_2}\right)^{-AAE_{BB}}, \tag{12}$$

The recombination of Eq. (10), Eq. (11) and Eq. (12) results in a 2-equation system with 4 unknowns. However, if the AAE

values of the evaluated sources are known or assumed, the equation system can be solved. One step further, when the

contributions of each of the sources to total absorption at $\lambda_2$ are known, converted to $eBC_{TR}$ and $eBC_{WB}$ concentrations through

Eq. (9), then the fractional contribution of $eBC_{BB}$ to total eBC ($eBC_{TOT}$) can be estimated as described by Zotter et al. (2017)

if the ratio of $MAC_{BB}(\lambda_2)/ MAC_{TR}(\lambda_2)$ is known or assumed:

$$\frac{eBC_{BB}}{eBC_{TOT}} = \frac{1}{1+\frac{MAC_{BB}(\lambda_2)}{MAC_{TR}(\lambda_2)} \cdot \frac{b_{abs,TR}(\lambda_2)}{b_{abs,BB}(\lambda_2)}}, \tag{13}$$

In the present study the assumption of $MAC_{BB}(\lambda_2)/ MAC_{TR}(\lambda_2)$ being~1 proposed by previous studies and tested by Zotter et

al. (2017) was kept. The pair of wavelengths chosen to apply in this method to apportion the contribution of local vehicular

emissions (also known as liquid fuel) from the regional agricultural biomass burning emissions (also known as solid fuel) were

470 and 950 nm. The representative source specific AAE were chosen as the 5th and 99th percentiles of the frequency

distribution of the hourly averages of the calculated AAE ($AAE_{TR}$= 0.85, $AAE_{BB}$=1.57) (Fig. S2 in the Supplement). Given

the observed large dispersion of the AAE values calculated at CHC-GAW, the urban values were extrapolated to the mountain

site for the bilinear source apportionment of BC.

### 2.5.2 Multilinear Linear Regression method (MLR) for apportionment of light absorption to PM segregated by sources

One of the limitations of the previously described Aethalometer method for source apportioning is that it only allows to separate

the contributions of two sources of aerosol particles to light absorption. Given that the contribution to $PM_{10}$ of dust and BB

increases during the dry season in both cities (Mardoñez et al., 2023), the aethalometer method cannot quantify nor separate

their influence from the influence of traffic. Moreover, the result of the source apportionment through the Aethalometer model

is highly sensitive to the choice of the representative AAEs. Furthermore, source specific AAEs are highly dependent on the

conditions under which combustion takes place and on chemical processing of the combustion emissions in the atmosphere.

Hence, the selection of AAEs needed in the previous method can be somewhat arbitrary, and extrapolating literature values

can significantly influence the results of the apportionment of BC. Thus, an alternative method was explored to calculate the

source specific AAEs, as well as to identify other possible sources of absorbing aerosol particles and to verify the results

obtained with the bilinear model.

Mardoñez et al. (2023) used the Positive Matrix Factorization tool (PMF v.5.0) to apportion the major sources of particulate

matter, based on the chemical speciation of the $PM_{10}$ filter samples collected during the first 15 months of the campaign at

both sites. The complete description of the factor analysis can be found in the mentioned article. Briefly, the temporal evolution



of 40 chemical species including PM, EC, OC, water soluble ions, metals and source-specific organic tracers and their associated uncertainties were used as input in the receptor model, following a multisite approach. A statistically stable solution was found for 11 factors assessed by the bootstrap and displacement methods, following the European source apportionment recommendations described in Belis et al., (2019). The complete description of the factor analysis can be found in Mardoñez
et al. (2023).

In the present study, a multilinear ordinary least-square regression (MLR) was performed to attribute the observed absorption coefficients ($b_{abs}$) to the PMF-resolved sources of $PM_{10}$. A similar approach was followed by Moschos et al. (2018) to apportion the sources of BrC in a rural and an urban background site in Switzerland. The calculated $b_{abs}$ at each of the 7-wavelengths were used as dependent variables and the PMF-resolved source contributions as explanatory variables in Eq. (14):

$$b_{abs}(\lambda)_{m,1} = (G_{m,n} \times \beta(\lambda)_{n,1}) + \varepsilon(\lambda)_{m,1}, \tag{14}$$

where $b_{abs}$ are the daily averaged absorption coefficients in $Mm^{-1}$ (starting at 9:00 to match the filter sampling time), $G_{m,n}$ are the STP mass contributions of the n sources for each of the m filters in $\mu g\ m^{-3}$, $\beta$ are the proportionality coefficients that represent the absorption efficiency of each of the sources in $m^2g^{-1}$, and $\varepsilon$ are the residuals that account for the difference between the observed and the modelled $b_{abs}$. The $\beta$ coefficients described above, similar to MAC values, mathematically allow
to relate $b_{abs}$ to the mass of the particles contained in each of the factors included in the analysis, thus describing their absorption efficiency as an ensemble of particles. However, a clear distinction must be made between the two quantities as they are conceptually different even though they share the same units. The MAC values represent an intrinsic physical property of a material (BC). Conversely, the $\beta$ proportionality coefficients found through the multilinear regression provide insight on the absorption efficiency per unit of mass of the collection of particles that constitute each factor (defined by the chemical profile
that defines each factor), which also includes non-absorbing material. To avoid further confusion, the symbol $\beta$ will be maintained when describing the absorption efficiency of the factors in the source apportionment of $b_{abs}$.

The uncertainties of the coefficients $\beta$ obtained by the MLR were estimated by bootstrapping the solutions 500 times, randomly selecting 70% of the datapoints each time to account for possible influence of extreme events. The median values of the estimated $\beta$ coefficients were then used to calculate the wavelength-dependent contributions to $b_{abs}$ of the individual sources.
Finally, the source-specific AAE were calculated using the power-law non-linear regression of the source-specific $MAC(\lambda)$ values obtained with respect to wavelength (Eq. 7).

Since the PMF analysis performed in Mardoñez et al. (2023) was done following a multisite approach, we chose to perform a multisite MLR in order to increase the number of data points included in the deconvolution. The hypothesis is therefore that BC presents the same optical properties at the two sites. Nevertheless, the contribution of each source to the measured mass of
BC will be presented separately per site.



## 3 Results

### 3.1 Observed BC concentration levels and variability

Table 2 displays the mean STP concentrations of eBC and EC measured in La Paz (LP), El Alto (EA) and Chacaltaya mountain station (CHC-GAW). EC concentrations were similar at both urban sites ($EC_{PM10-LP}$: 2.1±1.2 µg m$^{-3}$, $EC_{PM10-EA}$: 2.4±1.1 µg m$^{-3}$; $EC_{PM2.5-LP}$: 1.5±0.9 µg m$^{-3}$, $EC_{PM2.5-EA}$: 1.6±0.8 µg m$^{-3}$), whereas nocturnal EC concentrations in CHC-GAW were significantly lower (0.08±0.07 µg m$^{-3}$, rounded values of the concentrations reported by Moreno et al., (2024)). eBC mass concentrations were consistently lower than EC in La Paz and El Alto ($eBC_{LP}$: 1.1±1.2 µg m$^{-3}$, $eBC_{EA}$: 1.6±1.7 µg m$^{-3}$) but higher compared to EC in CHC-GAW ($eBC_{CHC-GAW}$: 0.2±0.3 µg m$^{-3}$). This implies that the $MAC_{AE,880nm}$ applied to derive eBC was too high for La Paz and El Alto and too low for Chacaltaya, when considering EC as the ground truth for BC mass, a finding confirmed in Section 3.2. Although the urban background eBC mass concentrations do not stand out as alarming, these concentrations can rapidly increase closer to the main roads as observed by Madueño et al (2020).

The CHC-GAW station is often influenced by urban emissions from LP and EA, as previously reported by Andrade et al (2015) and Wiedensohler et al (2018). These emissions get rapidly diluted on their way to CHC and arrive to the station with a lag of nearly three hours as a result of the diurnal PBL evolution above the Altiplano of the metropolitan area. The average concentrations of eBC measured at CHC-GAW are 12-15% of what is measured in the cities. However, during the hours when CHC-GAW is influenced by the mixing layer of the metropolitan area (10 - 16h), the concentrations in the mountain station are roughly 30% of what is measured in the city.

In comparison with the other two high-altitude Latin American cities where urban background concentrations of absorbing aerosol particles have been reported, $EC_{STP}$ concentrations measured in La Paz and El Alto (PM$_{10}$) are nearly two thirds of those reported in Bogota (Ramírez et al., 2018). Compared to Mexico City, the eBC mass concentrations in LP-EA are 37%-54% of the concentrations reported by Peralta et al. (2019) for 2016. As the latter study did not provide information on the temperature and pressure conditions under which the concentrations were measured, these percentages could potentially decrease to approximately 27%-40% if the eBC mass concentrations were reported under ambient conditions (using an annual mean temperature of 15.7 °C and a mean atmospheric pressure of 585 mmHg; Estrada et al., 2009; Hernández-Zenteno et al., 2002). Considering that vehicle emissions have been highlighted as the main source of EC and eBC in these three metropolitan areas, with vehicle fleets dominated by gasoline fueled vehicles, the observed concentrations of eBC and EC in La Paz and El Alto appear relatively elevated. This is particularly striking when considering that the combined population of the La Paz-El Alto conurbation is more than four times smaller than that of Bogota or Mexico City. This phenomenon may be attributed to various factors, including distinctions in population density, vehicle fleet density, combustion efficiencies, the unique topographical features of the cities, notably in La Paz, and pollutant dispersion efficiencies. However, a comprehensive examination of these factors lies outside the scope of the current study.

**Table 2. Annual and seasonal average concentrations of eBC from the Aethalometer and EC from TOA in La Paz (LP), El Alto (EA) and Mount Chacaltaya (CHC-GAW). EC concentrations in CHC-GAW were obtained from weekly nighttime (23h-8h) filters collected between April 2016 and August 2017.**





| mean ± sd [μg m⁻³] | | | | | | | | |
|---|---|---|---|---|---|---|---|---|
| *eBC* | | | | *EC (PM₁₀)* | | | *EC (PM₂.₅)* | |
| CHC-GAW | CHC-GAW (Nighttime: 23h-8h) | El Alto | La Paz | CHC-GAW (Nighttime: 23h-8h) | El Alto | La Paz | El Alto | La Paz |
| **annual** | 0.2±0.2 | 0.1±0.1 | 1.6±1.7 | 1.1±1.2 | 0.08±0.07 | 2.4±1.1 | 2.1±1.2 | 1.6±0.8 | 1.5±0.9 |
| **wet season (Dec-Mar)** | 0.1±0.2 | 0.1±0.1 | 1.0±0.8 | 0.8±0.9 | 0.04±0.02 | 1.6±0.8 | 1.5±0.7 | 1.4±0.7 | 1.2±0.6 |
| **wet-to-dry transition (Apr)** | 0.1±0.2 | 0.1±0.1 | 1.5±1.5 | 0.8±1.0 | 0.04±0.07 | 2.6±1.1 | 2.5±1.3 | 1.2±0.2 | 1.0±0.3 |
| **dry season (May-Aug)** | 0.2±0.2 | 0.1±0.1 | 1.9±2.0 | 1.5±1.6 | 0.13±0.09 | 2.9±0.8 | 3.0±1.2 | 2.1±0.8 | 2.2±1.1 |
| **dry-to-wet transition (Sep-Nov)** | 0.2±0.2 | 0.1±0.1 | 1.3±1.3 | 0.9±0.9 | 0.08±0.07 | 1.7±0.5 | 1.6±0.8 | 1.2±0.7 | 1.2±0.6 |


Equivalent black carbon mass concentrations (eBC) in La Paz and El Alto were strongly influenced by the local meteorology. Maximum concentrations are found during the dry season and the minimum concentrations take place during the wet season (Fig. 2.a). A significant decrease in concentrations can also be noted during the weekends compared to working days (Fig. 2.b), giving evidence of the presence of anthropogenic sources of BC. Figure 2 also displays an important difference between

the eBC concentrations measured at the urban sites compared to the mountain station. As can be deduced from Table 2, this difference increased by a factor two when EC concentrations of the urban/mountain sites are compared. This is partly due to the site specificity of the MAC values, as will be developed in the following sections. Nevertheless, caution should be made when comparing the average EC concentrations obtained from the 24-h filter samples collected in the urban sites and the ones obtained from the filter samples collected at nighttime in the GAW-CHC station.





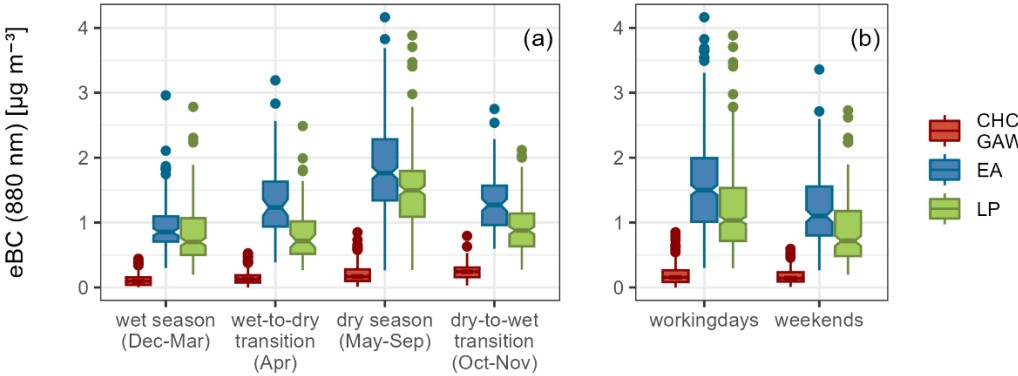


**Figure 2. Boxplots describing (a) the seasonal variation, and (b) the difference between working days and weekends of the daily average concentrations of eBC in La Paz, El Alto and Chacaltaya. The horizontal bar in the box represents the median value, the lower (upper) hinge represents the percentile 25 (75), and the lower (upper) whiskers represent the smallest (largest) observation greater (less) than or equal to lower (upper) hinge – (+) 1.5 times the interquartile range (IQR).**

In terms of diurnal variation, a clear bimodal pattern is observed throughout the seasons, which is characteristic of sites influenced by vehicular emissions (Fig. 3 (a)). This pattern has already been described by Wiedensohler et al. (2018) in a shorter temporal scale at EA station (during the transition period to the wet season in 2012). A less pronounced bimodal behavior was observed by the same study in the road site station in La Paz installed during the 2012 short campaign.

Maximum concentrations were observed during the morning rush-hour peak around 8:00 at both sites, followed by a rapid

decrease towards the minimum diurnal concentrations. This rapid decrease in concentrations results from the efficient ventilation and dilution caused by thermal convection, advection, and growth of the boundary layer. During the midday minimum, lower concentrations are achieved at EA compared to LP, which could be related to a better ventilation in El Alto due to the openness of the site compared to the sheer sided canyon where the city of La Paz lies. At CHC-GAW, concentrations only start to increase at 9:00, when the urban boundary layer has expanded enough to reach the altitude of the station, peaking

between 11:00 and 12:00. After mid-day, concentrations slowly decrease towards minimum concentrations during the evening. While the maximum diurnal concentrations of eBC are similar at EA and LP sites, the magnitude of the evening peaks constitutes the largest difference in the diurnal variation of eBC between these two sites. This second mode is associated to the evening rush-hour vehicular emissions coupled with the re-stratification of the atmosphere when at sunset, when temperature decreases. At EA, the evening peak reaches almost 60% of the morning peak's magnitude, whereas at LP the evening peak

represents only 40% of the morning peak's concentrations. Furthermore, the second mode in La Paz starts to steadily increase around 16:00 whereas in El Alto it increases more abruptly around 18:00. This delay in the start of the evening mode is likely linked to a combination of the earlier suset in the city of La Paz compared to El Alto (as a result of the difference in altitude and geography), and the delay in El Alto's evening traffic rush-hour due to the time it takes to commute back from La Paz to the city of El Alto.





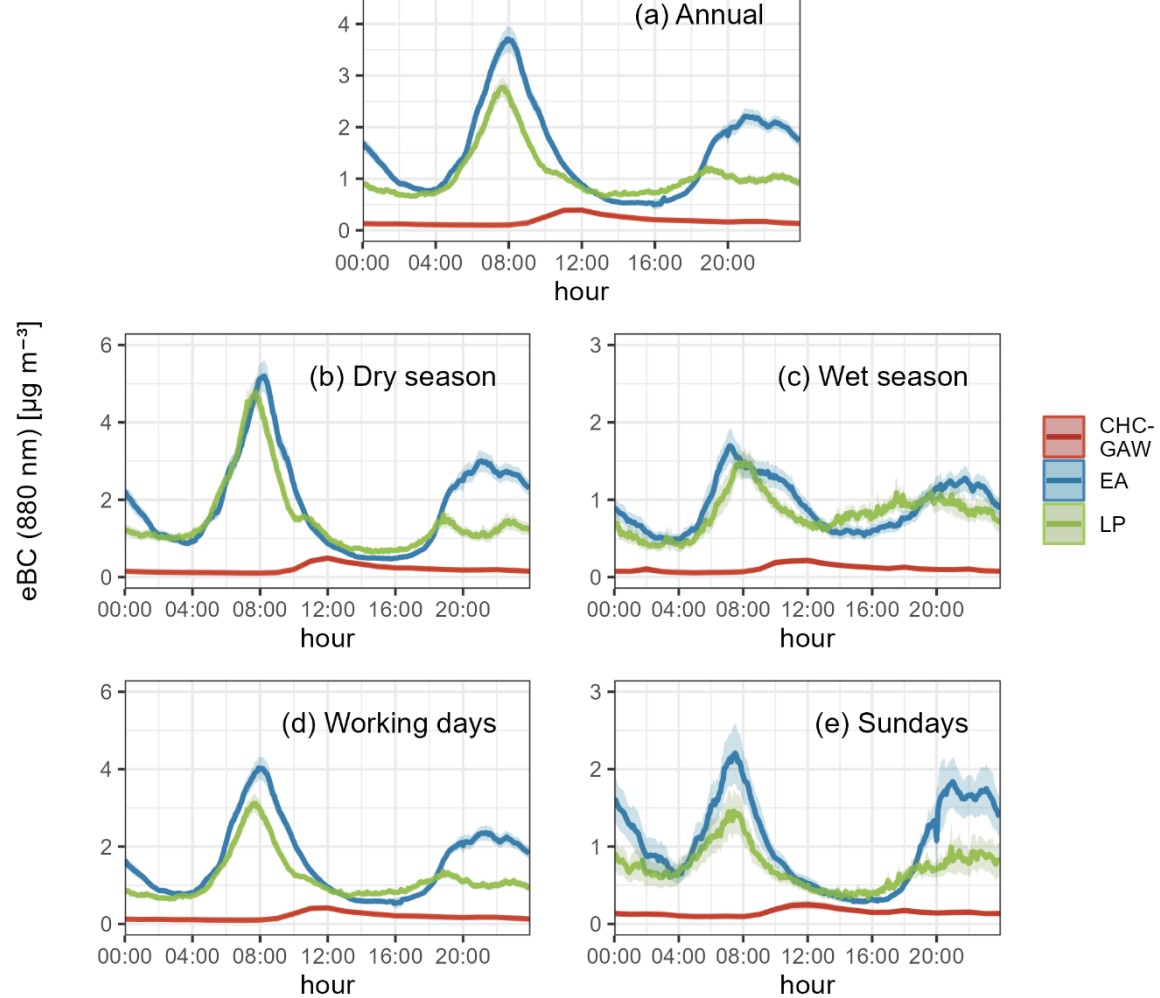

**Figure 3. Average diurnal variation of eBC concentration at the three sites during a) the complete sampling campaign, b) the dry season, c) the wet season, d) working days, and e) Sundays. The shaded areas around the mean values represent the 95% confidence interval. Note that the y-axes are scaled differently.**

Moreover, the evening mode in La Paz is composed of two smaller peaks: one occurring around 19:00 and the other around 23:00. This pattern is more pronounced during the dry season (austral winter) and is not discernible in El Alto (see Fig. 3 (b) and 3(d)). The first peak of the second mode of eBC average diurnal concentrations in La Paz seems to correspond to the evening rush hour, hence, it is not observed on Sundays (Fig. 3(e)). Instead, evening concentrations on Sundays steadily increase until 23:00 in La Paz. In contrast, during the wet season (austral summer), the second evening peak is no longer observable, likely due to the increase in night-time temperatures during this period.

A reduction in vehicular traffic (as it can be expected during weekends) as well as changes in meteorology (higher temperatures together with wet deposition) are both capable of reducing the average concentrations by a factor two at all sites. In addition,



the difference in magnitude between the morning and the evening peak is simultaneously modulated by the emitting sources and the meteorology. When the number of emitting sources is reduced, the difference between the morning and the evening peak is also reduced by 20% (Fig. 3(c) and 3(e)). Finally, the combinations of precipitation and higher temperatures makes the

evening peak concentrations similar at both sites, but also makes the morning peak in El Alto less symmetrical. A slight lump with maximum concentrations between 6:00 and 8:00 can be noted during the wet season in the average diurnal pattern. This pattern was also observed in Wiedensohler et al. (2018) both for eBC mass concentrations and particle number concentrations (PNC).

Wiedensohler et al. (2018) also reported eBC mass concentrations at the road-site in La Paz being on average 3 to over 10

times higher throughout the day than the ones registered at EA station, with a less pronounced bimodal diurnal pattern. In the present study, the average concentrations at LP were not significantly different than at EA, both in concentrations and diurnal variability, except for the difference in the magnitude of the evening peak. This demonstrates that an increase in the distance between the sampling site and the nearest road in La Paz, from < 10 m (Wiedensohler et al. 2018) to ~ 70 m horizontally and 40 m vertically (present study), is enough to significantly dilute traffic emissions and make the observations at the LP

background station comparable to the observations at the EA background station. Several previous studies, e.g. Alas et al., (2022) and Peters et al. (2014), have described similar rapid decreases in eBC mass concentrations between road and urban-background sites, as well as between traffic hotspots and less crowded routes located within a couple of hundred meters.

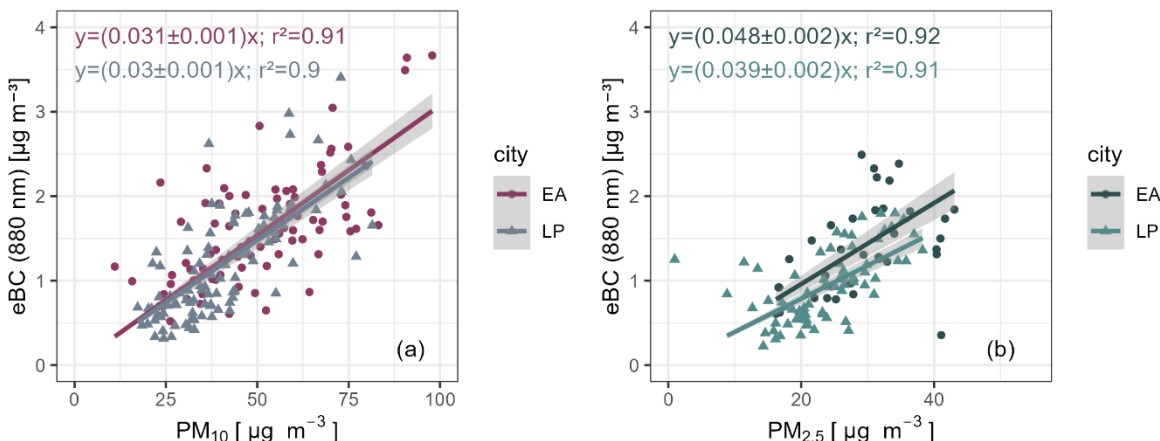

**Figure 4. Scatter plot of daily average eBC mass concentrations vs PM$_{10}$ (a) and PM$_{2.5}$ (b) concentrations at both sites (two outliers**
**were excluded from the ordinary least square regression displayed in the right panel).**

When compared to PM concentrations measured from daily samples collected during the same period, eBC displays similar trends as PM, with high dispersion around the linear fit (Fig. 4). At both sites and for both PM$_{10}$ and PM$_{2.5}$, eBC represents less than 5% of the total measured PM. Similar trends are observed when comparing EC and PM (Fig. S3 in the Supplement), with contributions of roughly 5 and 6% to total PM$_{10}$ and PM$_{2.5}$, respectively. Comparing the fraction of PM that corresponds

to eBC and EC in other high-altitude cities we observe that according to Ramírez et al., 2018b, the contribution of EC to PM$_{10}$



in Bogotá is almost 1.3 times higher (8.2%-9%) than in La Paz-El Alto. In contrast, in Mexico City, eBC contributions to $PM_{2.5}$ reported by Peralta et al. (2019), were almost 5 times higher than the ones observed in LP-EA (16%, calculated from average concentrations of eBC and $PM_{2.5}$).

## 3.2 Intrinsic aerosol and BC properties

The absorption Ångström exponent (AAE) and the Mass Absorption Coefficient of BC (MAC) of an aerosol sample are important aerosol parameters used for characterizing both source origin of BC, aging effects, and radiation impact. Here, we compute the variability of both parameters for the two cities using different methodologies according to Eq. (7) and Eq. (8). We also confronted such results with those from CHC under the specific conditions of advection from the metropolitan area, even though, as shown by Aliaga et al. (2021) air masses reaching CHC are typically a mix of different air masses.

### 3.2.1 Aerosol Ångström Exponent (AAE)

Average AAE coefficients of 1.1±0.2 were found for both La Paz and El Alto and of 1.0±0.3 in CHC-GAW. AAE values in CHC reach their minimum values during the wet season and their maximum between July and September, the period in which the biomass burning emissions reach their maximum contribution to the CHC (Moreno et al., 2024). During the day, lower AAEs are observed in CHC-GAW between 6:00 and 18:00, with a minimum around 10:00, when the urban mixing layer

develops enough to influence the mountain station and eBC mass concentrations reach their maximum (Fig. S4).

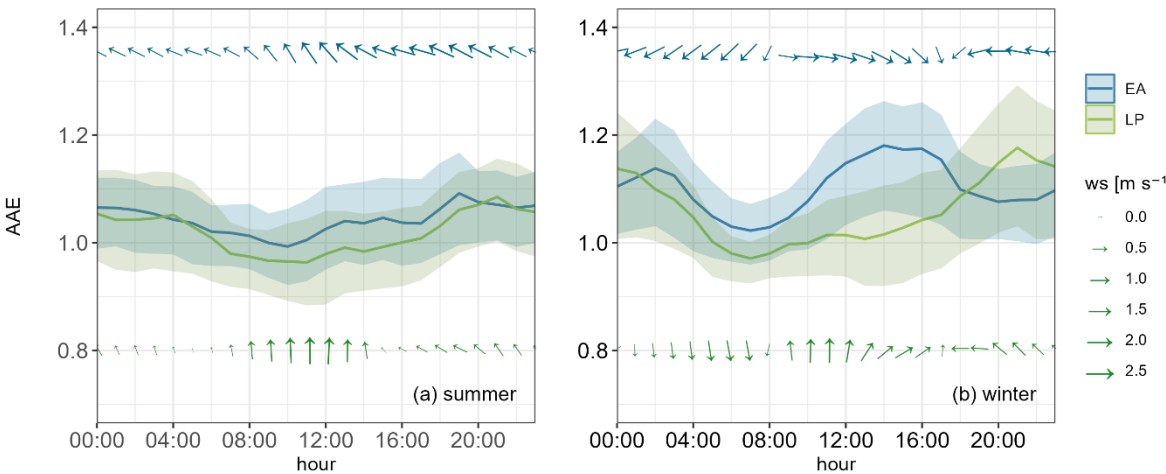

**Figure 5. Diurnal variation of the hourly median AAE (solid lines), and hourly average wind speed and wind direction (arrows above and below the solid lines, the arrowhead points to the direction to which the wind blows) at both sites during (a) winter and (b) summer. The shaded area behind the solid curves represents the 25 and 75 percentiles.**

The diurnal variability of AAE in the urban sites shows an overall decrease around the morning rush-hour peak towards the minimum values for both sites (Fig. 5). However, a difference in the evolution of this parameter throughout the day can be noticed between the sites during winter, the driest season of the year (Fig. 5(b)). A minor increment in the average AAE is



discernible exclusively in winter at El Alto spanning from 9:00 to 18:00 (Fig. 5.b), period in which the average wind direction changes in El Alto from being mostly easterly/north-easterly to westerly/north-westerly. This change in the wind direction

seems to allow the incursion of particles with different optical properties coming from the altiplano, e.g. dust. This change in the AAE during this period of the day takes place when eBC mass concentrations reach their minimum, thus, no significant impact in absorption was observed at 880 nm. However, this reveals the potential influence of additional sources of absorbing particles in El Alto besides traffic and biomass burning. Further corroborating this phenomenon in Fig. S5, in the Supplement, winter days are categorized according to their dust factor influence, which was resolved through the Positive Matrix

Factorization (PMF) technique as described in Mardoñez et al. (2023). Marginally higher AAE values are discernible during daylight hours on winter-days that are characterized by dust factor mass concentrations exceeding the 0.75 percentile threshold (7 days in total). This trend also coincides with elevated wind speeds and changing wind direction. Conversely, winter days marked by dust factor contributions below the 0.25 percentile threshold (10 days in total) coincide with non-changing wind patterns and a less pronounced diurnal variation.

In La Paz, winds are practically bidirectional throughout the year due to the topography of the city, winds typically follow the direction of the canyon, blowing mainly from the south. No change in the AAE diurnal pattern is observed at this site besides the expected decrease of AAE during daytime resulting from fresh vehicular emissions.

**3.2.2 Mass Absorption Cross-section (MAC)**

The comparison of the daily averaged eBC mass concentrations to the measured EC concentrations obtained from the filter

samples yielded good spearman correlations for all sampling sites (r≥0.7), especially in La Paz (r≥0.9) (Table S1 in the Supplement). The fitting parameters obtained from the least-squares regressions performed among both variables are very similar for both urban sites and for both size cuts with slopes between 0.49-0.54 and offsets between 0.15 and 0.32 µg m$^{-3}$. In contrast, the weekly averaged eBC mass concentrations at CHC-GAW showed to be 1.7 times larger than the EC concentrations measured from the simultaneously collected filter samples, leading to outstandingly high MAC values (Table

3). As previously mentioned in section 3.1, this contrast shows that the MAC$_{AE,880nm}$ used to calculate eBC mass concentrations was not representative of the actual optical properties of the sampled aerosol particles, i.e. it overestimated the mixing state of the urban aerosol, whereas it underestimated the mixing state of the aerosol particles sampled in CHC-GAW.

The average ratio of the calculated absorption coefficients b$_{abs}$ and EC mass concentration (using a C$_f$=2.78 for AE33 and C$_F$=3.08 for AE31), resulted in relatively low MAC$_{EC}$ at the urban sites (Table 3). The average of the daily MAC$_{EC}$ values

found at both cities La Paz and El Alto are within those reported for urban and urban background areas where the absorbing particles had already undergone some mixing/aging processes (Kondo et al., 2009; You et al., 2016; Chen et al., 2017; Cui et al., 2016). Despite intense radiation, the ageing process does not appear to be faster in the high altitude as respect to other places. Average MAC$_{rBC}$ values obtained from the hourly ratios of absorption coefficients over rBC mass concentrations were similar to the MAC$_{EC}$ for EA, but much higher values were obtained for LP. The latter result indeed comes from an instrumental

limitation. The detection capability of SP2 outside the size range 80-650 nm is very limited (Pileci et al., 2021) thus the rBC



mass is typically underestimated when the rBC mass distribution peaks below the 80 nm lower detection limit of the SP2-XR. This was the case for 90% of samples in La Paz, while in El Alto, rBC mass distributions were normally monomodal with a peak around 180 - 200 nm.

**Table 3. Average $MAC_{EC}$ and $MAC_{rBC}$ calculated following Eq. (8) using EC and rBC mass concentrations measured at the three sampling sites. The MAC values were extrapolated to other commonly reported wavelengths using the corresponding daily mean AAEs (with average values of 1.1±0.2 in the urban area and 1.0±0.3 in CHC-GAW).**

| BC mass | | $\lambda$ [nm] | CHC-GAW[i] Nighttime (23h-8h) | CHC-GAW Daytime (10h-16h) | EA | LP[ii] |
|---|---|---|---|---|---|---|
| *EC-PM$_{10}$* | **$MAC_{EC}$** | 550 | 26.0±11.2 | --- | 9.0±2.2 | 7.4 ±2.7 |
| | ± | 637 | 21.4±9.3 | --- | 7.7±1.9 | 6.6±2.4 |
| | **SD [m$^2$ g$^{-1}$]** | 880 | 14.8±6.9 | --- | 5.4±1.3 | 4.5±1.7 |
| *EC-PM$_{2.5}$* | **$MAC_{EC}$** | 550 | --- | --- | 9.3±3.7 | 8.6±1.8 |
| | ± | 637 | --- | --- | 8.2±3.1 | 7.6±1.7 |
| | **SD [m$^2$ g$^{-1}$]** | 880 | --- | --- | 5.8±2.2 | 5.1±1.1 |
| *SP2-XR (SP2-C in CHC)* | **$MAC_{rBC}$[iii]** | 550 | 28.3±8.5 | 22.6±4.9 | 8.8±2.3 | --- |
| | ± | 637 | 24.4±7.3 | 19.5±4.2 | 8.1±2.1 | --- |
| | **SD [m$^2$ g$^{-1}$]** | 880 | 17.7±5.3 | 14.1±3.0 | 5.6±1.2 | --- |

[i] $MAC_{EC}$ in CHC-GAW are calculated only during night-time periods (23:00-8:00) between April 2016 and April, when the station is outside the direct influence of the urban mixing layer. $MAC_{rBC}$ in CHC-GAW calculated during the period in which the station is under the direct influence of the urban mixing layer (10:00-16:00) between April and May 2018.

[ii] Median $MAC_{rBC}$ in La Paz were calculated for the rBC mass concentrations that resulted from rBC mass-size distributions that peaked above the detection limit of the SP2_XR (i.e. approx. 10% of the cases) and can be found in Table S2, in the Supplement.

[iii] MACrBC values for CHC-GAW obtained from Renzi et al. (in prep) and extrapolated to 550 nm and 880 nm through Eq. (7) assuming a constant AAE=1 that corresponds to the average AAE in CHC.

From Table 3 it can be observed that despite the EC and rBC concentrations having been measured during different periods of time, with different measuring techniques and different time resolutions, average $MAC_{EC}$ and $MAC_{rBC}$ remain consistent in EA and comparable in CHC. Nevertheless, the spectral dependence of $MAC_{EC}$ and $MAC_{rBC}$ values is different. It should be noted that the periods in which they are calculated are also different. Therefore, caution should be taken not to over-interpret the small differences encountered between $MAC_{EC}$ and $MAC_{rBC}$ values.



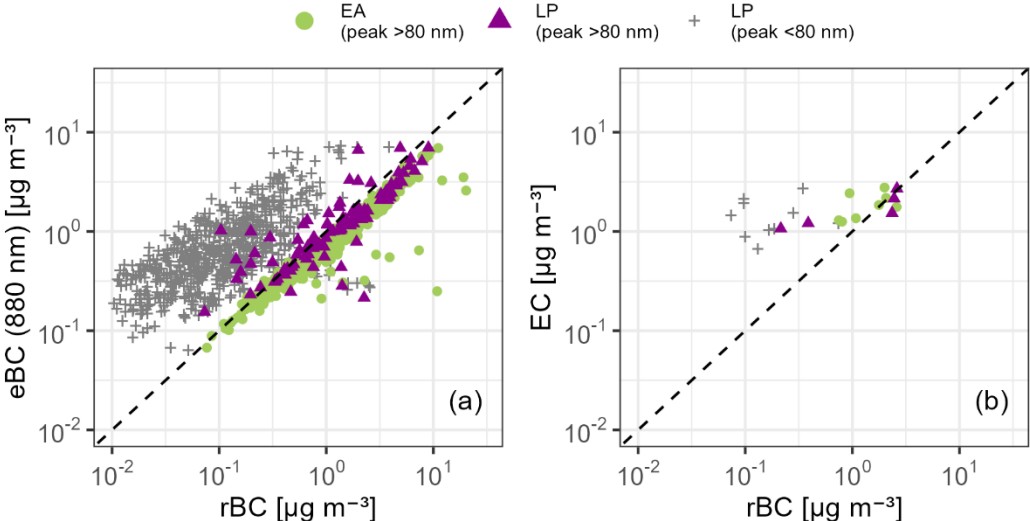

**Figure 6. (a) Hourly average eBC and (b) daily average EC mass concentrations vs rBC concentrations measured during Apr-May 2018 at both sites. Color and shape distinguish mass size distributions of rBC that peaked within (LP: purple triangles and EA: green circles; corrected) or outside the size range of the SP2-XR (gray crosses; non corrected).**

Figure 6 displays the comparison of the simultaneous integrated mass concentrations of rBC compared to the optically estimated eBC and thermally/optically derived EC mass concentrations sampled in LP and EA between April and May 2018. In color are displayed the cases when in LP (purple triangles) and in El Alto (green circles) the mass size distributions peaked above the SP2 detection limit (>80 nm), which account for the undetected mass of the particles in the size range outside the detection limit of the instrument through a lognormal fit of the rBC mass-size distribution. In gray (crosses) are the cases in LP where the distribution peaked below 80 nm, which cannot be accurately corrected for undetected mass. In the latter case, the rBC mass concentrations are highly underestimated, accounting for less than 15% of the mass of absorbing aerosol. Reliable $MAC_{rBC}$ coefficients can therefore not be calculated for LP due to the large fraction of the mass missed, and no evident relation between meteorological parameters and the particle's aging state was observed in La Paz associated to these measurements. Nevertheless, this points out to major differences in the properties of BC between the two sites that will be further discussed in Modini et al. (in prep). In the context of the present study, we restricted the calculation of $MAC_{rBC}$ to the cases when the size distribution of rBC peaked within the detection limit of the SP2-XR (Table 3). Given the low number of datapoints that meet this criterion in LP, mean values of $MAC_{rBC}$ are strongly influenced by the few extreme events, resulting in higher $MAC_{rBC}$ values. Median values are closer to the $MAC_{EC}$ and $MAC_{rBC}$ observed at EA (Table S2 in the Supplement). No substantial variability in the $MAC_{rBC}$ values was observed on hourly timescales in EA. This indicates that the sources of emissions and the pathways through which they were transported to the EA site were relatively constant throughout the day and throughout the measurement period. Nevertheless, two minima are observed in the diurnal variation of $MAC_{rBC}$ at EA, which correspond to the morning and evening rush-hour peaks (Fig. S6 in the Supplement). In contrast, the maximum $MAC_{rBC}$ values observed throughout the day take place in the early morning and at midday. This provides evidence that fresh BC from




traffic has low MAC, whereas aged BC has somewhat higher MAC, likely due to an increased degree of internal mixing. The diurnal variability in LP is not presented due to low statistical power.

Table 3 also displays for CHC-GAW the average nighttime $MAC_{EC}$ values calculated for the period between April 2016 and August 2017 (when weekly nighttime $PM_{10}$ filter samples were collected), and the average $MAC_{rBC}$ values calculated for the period between April and May 2018 on an hourly basis during the hours in which the station is under the influence of the urban boundary layer (10-16 h) and during nighttime (23h-8h). It is possible to observe that the average (Table 3) and median (Table S2 in the Supplement) MAC values obtained from these two very different techniques for measuring rBC/EC mass

concentrations, during different periods of the year and during different periods of the day, are comparable as highlighted by Figure 6 and quantitavely expressed in Table S2. However, they are outstandingly high compared to other European regional background stations, which according to Zanatta et al. (2016) are in average 10.0 $m^2g^{-1}$. The reasons behind this very high MAC values are further discussed in Renzi et al. (in prep.). Although relatively high MAC values have been observed in other stations, like Puy de Dôme (seasonal average at 637nm: 13.4±1.6-19.9±1.7, Zanatta et al., 2016), instrumental artifacts in the

filter based measurements, resulting in very high MAC values, cannot be ruled out. A deepened discussion on this can be found at Renzi et al. (in prep.).

## 3.3 Intrinsic aerosol and BC properties

### 3.3.1 Aethalometer model

The apportionment of the contribution of vehicular emissions and agricultural biomass burning emissions to absorption was

calculated following the methodology described in section 2.5.1. Average contributions attributed to BB emissions present highest values during the BB season (Jul-Nov) and minimum values during the beginning of the wet season (Dec-Jan). The median contribution of BB to eBC mass concentrations based on the Aethalometer method was estimated to be 29% at CHC-GAW and 25% at the urban sites during the BB season (Fig. 7), and it decreased to 20%, 16% and 11% in LP, EA, and CHC-GAW, respectively, outside this period. Extremely low eBC mass concentrations measured at the CHC mountain station make

the apportioned BB contributions to eBC to be widely spread around the median values in CHC-GAW. Even though, the percentage contributions are consistent among the three sites during the biomass burning season, the contributions of BB between December and June appear rather high since outside the biomass burning season little to no agricultural burning of crops is expected. This result is likely biased by the selection of $AE_{TR}$ and $AE_{BB}$ as will be shown in section 3.3.2. In Fig. S7 in the Supplement, the resulting time series of the apportioned eBC mass concentrations attributed to biomass burning practices

are shown, which display similar seasonal trends throughout the years at the three sites.



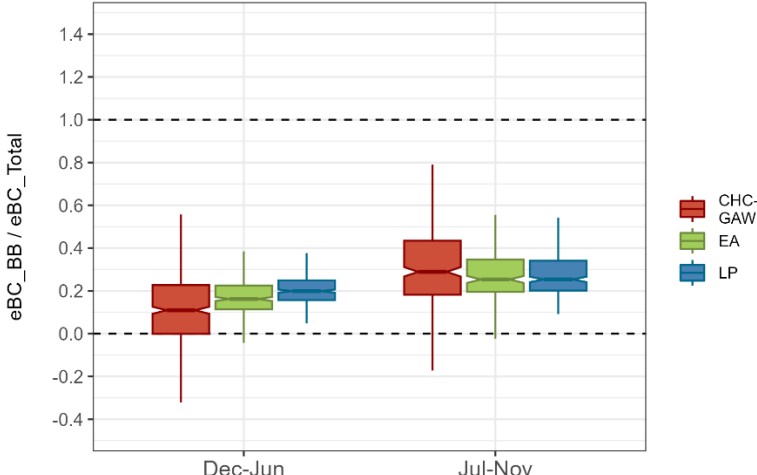

**Figure 7. Box whiskers plot of the daily eBC_BB/eBC$_{TOT}$ at 950 nm resulting from the bilinear Aethalometer method displaying maximum contributions during the biomass burning season (Jul-Nov). The horizontal bar in each box represents the median value. The lower (upper) hinge represents the percentile 25 (75), and the lower (upper) whiskers represent the smallest (largest) observation greater (less) than or equal to lower (upper) hinge – (+) 1.5 times the interquartile range (IQR).**

### 3.3.2 Estimation of source specific absorption efficiencies and AAE through multilinear regression

One of the limitations of the Aethalometer bilinear method is that it only allows the differentiation of two emission sources with significantly different optical properties, which might be suited for many cities in OECD countries. However, in La Paz/El Alto, this assumption does not necessarily hold, as other sources of BC may be impacting the urban air quality. Hence, we used the results from Mardoñez et al. (2023) where an integrated PMF approach was employed and 11 different sources of PM$_{10}$ were identified.

Briefly, out of the 11 resolved sources, dust and the ensemble of vehicular emissions (Traffic 1, traffic 2, lubricant oil, non-exhaust emissions) were found to be responsible for almost 50% of the measured mass concentrations of PM$_{10}$. The overall contribution of agricultural BB was ~8% with maximum contributions between June and September (during the biomass burning season that takes place during the dry season). Two other factors associated to secondary aerosol formation (secondary sulfate and secondary nitrate) were responsible for nearly 15% of PM$_{10}$ mass on a yearly basis. However, the chemical profiles of both secondary factors evidenced an influence of traffic emissions by the presence of some species that are tracers for vehicular emissions, such as EC (in secondary nitrate), OC and alkanes. This suggested that the gaseous precursors of the tracers for the secondary factors (sulfate, nitrate, and ammonium) could be at least partly co-emitted with primary traffic emissions, and therefore the PMF is not able to fully separate them. A small factor in terms of percentage mass contribution was found to be associated to open waste burning predominantly in El Alto between May and August. Finally, the natural sources of aerosol particles accounted for the remaining ~17% of PM$_{10}$ (Mardoñez et al., 2023).

In the present study, only the sources that contributed positively and that presented a p-value<0.05 at multiple wavelengths were included in Eq. (14) (section 2.5.2) for the analysis (i.e. biomass burning (BB), secondary-nitrate, traffic 1 (TR1), traffic



2 (TR2), waste burning). Although some evidence of the influence of dust was observed in the AAE in EA, the dust factor was not considered in the source apportionment since the p-value for this source was only significant for the shortest wavelength (370 nm). The natural primary and secondary factors did not significantly contribute to absorption, and neither did the factors associated to lubricant or non-exhaust emissions.

The regressions displayed correlations $R^2 \geq 0.78$ for all wavelengths, with a reconstruction > 80% of the observed absorption
coefficients. The obtained source-specific β coefficients (absorption per mass unit of each source) are presented in Fig. 8. Open waste burning stands out as the most efficient source in terms of absorption, followed by sources directly or indirectly linked to traffic (TR1, TR2, secondary nitrate), and biomass burning.

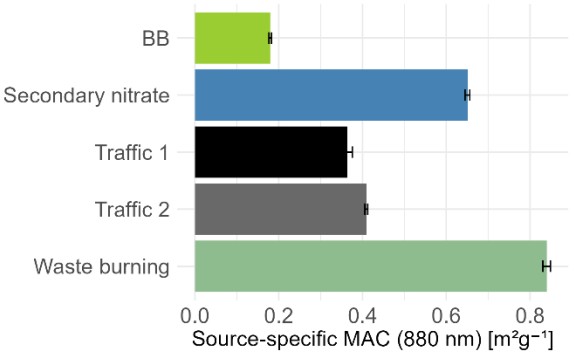

**Figure 8. Median source-specific mass absorption cross sections (ß) obtained from the multisite MLR between b$_{abs}$ (880 nm) and the**
**mass contribution of the six sources of PM$_{10}$, resolved by Mardoñez et al. (2023), that significantly contributed to the measured absorption coefficients. The black error bars represent the 95% CI of the median of the 500 bootstrap runs.**

In terms of their contribution to absorption at 880 nm (Fig. 9), i.e. source-specific β x mass concentration of the factor, TR2, waste burning, and secondary nitrate stand out as the major sources of absorbing aerosols at both sites. It is important to highlight that the chemical profile of the secondary nitrate factor contained 10-20% of the total measured EC and OC, together
with some alkanes. These components, tracers of traffic emissions, were not successfully separated from the tracers of secondary nitrate (ammonium and nitrate), thus, revealing its association to primary vehicular emissions. The chemical profile of the sources included in the MLR model can be found in Fig. S8 in the Supplement as well as the percentage mean contribution of each source factor to the total mean concentrations of EC and OC (Table S3).

Having traffic and traffic related factors as the major sources responsible for absorbing aerosol particles is coherent with the
results obtained by the aethalometer method, and with the mean AAEs~1 obtained at each site. However, at EA open waste burning becomes a dominant factor in terms of absorption, which only comes as a fourth in the sources ranking at LP. As stated by Mardoñez et al. (2023), open waste burning is not the institutional method for waste management in the urban areas, for which it is likely that these local emissions could originate from punctual sources of waste burning, or from the emissions of industrial and open commercial areas in El Alto. Regional BB comes as the fifth source at EA and fourth at LP (tying with
TR1 and waste burning). It is noteworthy that open waste burning represented less than 5% of the total PM$_{10}$ mass concentrations at both sites (Mardoñez et al. 2023), however, it becomes one of the leading sources contributing to absorption





in EA. Figure S9 in the Supplement displays the absorption contribution of each of the sources at 470 nm in which the ranking of the sources is maintained for EA, whereas in LP biomass burning becomes the third most important source in terms of absorption.

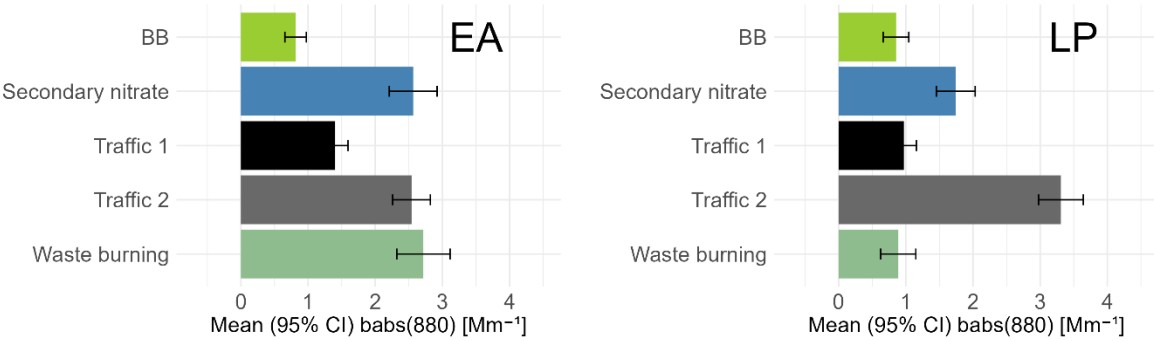


**Figure 9. Mean contribution to absorption at a wavelength of 880 nm of the five sources of PM$_{10}$ resolved by Mardoñez et al. (2023) (left: EA, right: LP) that presented a positive contribution to total absorption and a p-value <0.05 in the multilinear ordinary least-squares regression.**

Moreover, Table 4 presents the mean source specific AAE calculated, as described in Section 2.5.2, including all seven
wavelengths as well as excluding the shortest one (370 nm). The highest (>> 1) AAE using the pair of wavelengths found corresponds to BB, which is at the lower end of the reported literature values, which is well explained by the co-emitted BrC. However, AAE for BB including 370 nm is closer to the values recommended by Zotter et al. (2017) for wood burning. The AAE value obtained excluding the shortest wavelength confirms the AAE$_{BB}$ selected for the Aethalometer method in section 2.5.1 and is comparable to the lower end of AAE values found by Rizzo et al. (2011) during the BB season in a Brazilian
pasture site located in the Amazon. The remaining sources, influenced by local anthropogenic activities including vehicular emissions and open waste burning present an AAE close to 1, corresponding to freshly emitted, relatively pure, BC particles. The uniformity of AAE among sources, excluding biomass burning (BB), underscores the limitation of inferring additional source information through the incorporation of three or more sources and the consideration of three or more wavelengths in an advanced Aethalometer model. Consequently, there is a necessity for independently constrained source contributions when
estimating aerosol mass.

The presence of EC in the chemical profile of the open waste burning factor, as described in Mardoñez et al. (2023), and the AAE=1.1-1.2 associated to it suggest that this source could result from high temperature combustion processes. Furthermore, profile TR2 presents a lower AAE than TR1. Given that TR1 and TR2 presented different chemical profiles in Mardoñez et al. (2023), this difference in the wavelength dependency of their contributions to absorption could also give insights of a
difference in the nature or the state of the absorbing aerosol particles emitted by these two sources. Based on the experiments of Schnaiter et al. (2005), Liu et al. (2018) has estimated AAE<1 for aged diesel BC particles (compact and coated). Other European studies have also reported low AAE in experiments that involved diesel buses (accelerating) (Helin et al., 2021) or diesel dominated regions (Zotter et al., 2017). Nevertheless, given that the vehicle fleet, the fuel, and the combustion conditions





in LP-EA are different than the ones in the mentioned studies, only an in-situ characterization of the exhaust emissions could
unveil the identity of each factor. The pairwise source specific AAE calculated following the method used by Zotter et al.
(2017) are displayed in Table S4, showing very similar results to those in Table 4, except for BB and waste burning. Waste
burning AAEs were slightly lower compared to the nonlinear least-squares estimates, but closer to the values reported by
Zotter et al. (2017).

**Table 4. Source-specific AAE obtained from the non-linear regression of the source-specific mass absorption MAC (λ) cross-sections**
**and wavelength.**

|  | AAE (370 - 950 nm) | AAE (470 - 950 nm) |
|---|---|---|
| **BIOMASS BURNING** | 1.75±0.08 | 1.48±0.03 |
| **TRAFFIC 1** | 1.02±0.05 | 0.93±0.06 |
| **TRAFFIC 2** | 0.81±0.03 | 0.83±0.04 |
| **SECONDARY NITRATE** | 0.98±0.01 | 0.99±0.02 |
| **WASTE BURNING** | 1.17±0.03 | 1.12±0.05 |

The time series of the daily contribution of BB to absorption obtained by the two methods of BC source apportionment, MLR
and Aethalometer method, are presented in Fig. S7. The apportioned absorption coefficients calculated through the
Aethalometer method are clearly higher than those obtained from the MLR. However, when $AAE_{TR}$ in the Aethalometer
method is increased to 1 (the average AAE of the traffic and traffic influenced factors) the apportioned absorption coefficients
attributed to biomass burning obtained from both methods are brought closer together (Fig. 10). This change in the predefined
$AAE_{TR}$ decreased the previously calculated median contribution of BB to absorption from 29% to 17% at CHC-GAW and
from 25% to 12% at the urban sites during the BB season and resulted in a general agreement between both absorption
apportioning methods at both sites (except for the maximum contributions obtained by the Aethalometer method at EA in
2016, and in March 2017 in La Paz). These discrepancies could be influenced by the presence of other confluent sources that
have AAE>1 and an increase of their contributions during the same periods. Nevertheless, it is possible to observe that the
efficiency of predicting low contributions of BB by the Aethalometer method results in noisy absorption coefficients below 0.



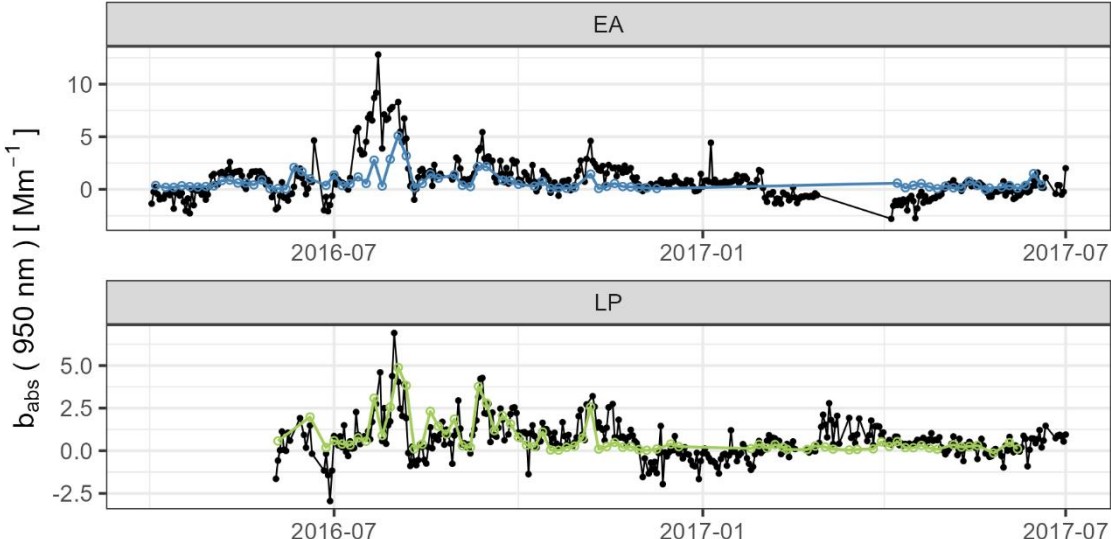

**Figure 10. Timeseries of the contribution of agricultural biomass burning to the absorption coefficients at 950 nm measured at the two urban background sites (LP and EA) using two different apportioning methods: Aethalometer method (black lines, using AAE$_{TR}$=1 and AAE$_{BB}$=1.5, percentiles (1,99)) and MLR deconvolution from the mass contribution of PM$_{10}$ sources resolved by Mardoñez et al. (2023) (colored lines).**

We can conclude that both source apportionment methods highlight local anthropogenic activities as the main sources responsible for the observed absorption, dominated by sources influenced by primary and secondary vehicular emissions. Nevertheless, the aethalometer method is sensitive to the assumed values of AAE$_{TR}$ and AAE$_{BB}$. The contributions of coexisting sources with similar optical properties such as TR1, TR2 and waste burning can be masked in the two-source aethalometer model, partially leading to an overestimation of the contribution of BB to total absorption. In addition, BB is responsible for a small but significant fraction of the observed absorption despite its emitting sources being located hundreds of kilometers away from the urban centers to the other side of the Andes. Despite the large contribution of dust to PM mass concentrations, the chemical and optical properties of dust make its contribution to total absorption almost insignificant.

**4 Conclusions**

The average background concentrations of eBC in La Paz (LP) and El Alto (EA) are comparable amongst the sites and lower than the concentrations reported for other high-altitude Latin-American megacities. The different local meteorological conditions make the concentrations of eBC in EA much higher during the evening compared to LP. A factor of two of difference in magnitude can be observed between working and non-working days at both sites, as well as between the dry and the wet season, indicating the important role that weekly anthropogenic activities and meteorology play in modulating the eBC mass concentrations. The influence of the conurbation can also be observed at the global station CHC-GAW, located ~20 km



away, showing concentrations that are roughly 35% of what is measured simultaneously at the urban area (within the hours CHC-GAW is influenced by the urban mixing layer, ~10-16h).

Despite the specific conditions of the urban sites, the intrinsic properties of BC are not fundamentally different than at other
places. The MAC values and the AAE estimated for both cities show that at an urban background level, eBC mass concentrations are dominated by relatively fresh vehicular emissions that do not undergo drastic ageing processes. Nevertheless, the peaking at low diameters of the BC mass size distributions in the city of La Paz is a phenomenon that is not expected at an urban background site and remains to be further investigated.

The main sources of absorbing particulate matter in LP-EA are rather local. Vehicular emissions are the first target to tackle
from an air quality perspective as well as to reduce the impact these emissions can have on climate. Other sources contributing to emitting BC are open waste burning (particularly in EA) and regional agricultural biomass burning. This is noteworthy since open waste burning occurs in many cities in developing countries, in Bolivia and elsewhere.

It was observed that the Aethalometer method can overestimate the contributions of biomass burning in the presence of a third source of absorbing aerosol particles, such as open waste burning. The multilinear regression allowed for the estimation of the
source-specific MAC values, the source contribution to total absorption and the source-specific absorption Ångström exponent (AAE) of the sources directly or indirectly associated to vehicular emissions, biomass burning and open waste burning.

Rigorous policies controlling the open waste burning and the size/state of the circulating vehicle fleet are therefore imperative to reduce the impact of BC on climate and on health of the inhabitants of the conurbation. Furthermore, the detection of ultrafine BC (Black Carbon) particles with exceptionally small diameters at an urban background site is a phenomenon that
requires further investigation since it represents a potential higher risk of exposure to ultrafine particles of the local population.

## Code availability

All calculations were performed in R version 4.3.2 and the code is available upon request.

## Authors contribution

MA, PL, AW, GM, MP, AM, GU, JLJ, RK, FV, IM, and IM participated in the conceptualization of the experimental setup
and design. VM participated in the data curation. VM, GM, MP, RM, and LR participated in the formal analysis and the development of the methodology. GU, MA, PL, JLJ, AA, RK, AM, MGB and PG were involved in the funding and resource acquisition. PL, GM, MP, MA, GU helped with mentoring, supervision, and validation of the methodology, techniques, and results. VM was responsible for the data processing and the writing of the original manuscript draft. PL and GU revised the original draft. All the co-authors reviewed and edited the paper.



**Competing interests**

The authors have the following competing interests: Griša Močnik is employed by Haze Instruments d.o.o., a manufacturer of the aerosol instrumentation. Droplet Measurement Techniques (DMT) provided an SP2-XR for the additional measurements in Pipiripi during the 2018 field campaign. Furthermore, at least one of the (co-)authors is a member of the editorial board of Atmospheric Chemistry and Physics

**Acknowledgements**

The authors wish to thank all the people from the different laboratories (LFA, Idaea-CSIC, IGE, Air O Sol analytical platform) who actively contributed over the years in filter sampling and/or analysis. The engineer Celine Voiron who performed the OC/EC analysis is warmly recognized. Sergio Rios and Erika Miranda (deceased) of GAMLP (Gobierno Autónomo Municipal de La Paz) who provided access and facilitated tasks at Pipiripi; IIF personnel that helped in logistics during the campaign; Undergrad students who collected samples: Yaneth Laura, Grover Salvatierra, Manuel Roca, Dayana Calasich, Ever Huanca, Zarella Tuco, Santiago Herrera, Mónica Vicente, Monserrat Zapata, and Roxana Copa.

**Financial support**

This research has been supported by the Institute de Recherche pour le Développement (IRD) France and IRD delegation in Bolivia, by Observatoire de Sciences de l'Univers de Grenoble (OSUG) partly through Labex OSUG@2020 (ANR10 LABX56), by CNRS/INSU and Ministère de l'Enseignement Supérieur et de la Recherche with contribution to ACTRIS-FR and SNO-CLAP, Slovenian research agency ARIS program P1-0385, by EU H2020 MSCA-RISE project PAPILA (Grant #: 777544) and by in-kind fundings by GU and JLJ. The PhD scholarship of Valeria Mardoñez was funded by the ARTS program of the Institute de Recherche pour le Développement (IRD) France. PSI's contribution received support from the ERC under grant ERC-CoG-615922-BLACARAT. The EC funded project EUROCHAMP-2020 (grant no.: 730997; now integrated in the ACTRIS ERIC) provided TNA support for AM to access the CCSM for training on and preparation of the SP2 for this campaign. LR received training at CCSM on SP2 data analysis, financed by Bologna University PhD program.

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
