# Peer review of "Atmospheric Black Carbon in the metropolitan area of La Paz and El Alto, Bolivia: concentration levels and emission sources"

_EGUsphere, 2024_

## Author Comment (AC1)

The authors would like to thank the anonymous referee # 2 for taking the time to review the manuscript and for considering our study suitable for publication in ACP. We thank them for the suggested bibliography that allowed us to improve the discussion section of the manuscript. We hope that they'll find our responses to their comments adequate.

Below you will find the list of the referees' observations (bold), followed right after by the author's responses (normal font) and the respective changes made to the manuscript (italic), highlighting the sections that were modified.

**1. Some studies have shown that the Single Particle Soot Photometer, when relying on scattering measurements to determine optical particle size and mixing state, can be influenced by microphysical characteristics. Please refer to relevant explanations for further clarification.**

**References**

**Wu, Y., Cheng, T., Zheng, L., Zhang, Y., and Zhang, L.: Particle size amplification of black carbon by scattering measurement due to morphology diversity, Environmental Research Letters, 18, 024 011, https://doi.org/10.1088/1748-9326/acaede, 2023.**

**Luo, J., Hu, M., Qiu, J., Li, K., He, H., Sun, Y., and Geng, X.: Technical note: Numerical quantification of the mixing states of partially-coated black carbon based on the single-particle soot photometer: Implication for global radiative forcing, EGUsphere [preprint], https://doi.org/10.5194/egusphere-2024-1155, 2024.**

We thank the reviewer for pointing out the limitations of the diameters of BC-containing particles derived from the SP2 measurements. They indeed need to be considered when discussing the size distributions inferred from the scattering signals measured by the SP2. However, the analysis of the size distributions of coated particles and their scattering signal was reserved for a separate article focused on the microphysical properties of BC, which is currently in preparation. In the present study, we limited to analyzing the mass concentrations obtained from the SP2 (that rely in the incandescence signal emitted by the cores of BC) and their comparison to the absorption coefficients associated to them. The classification of mass concentrations based on the mean mass equivalent core diameters in Figure 6 is based on the peaking of the mass size distribution, which is estimated from the incandescence signal.

**2. Some studies have also found that the Aethalometer model, when used for tracing the sources of black carbon, can be influenced by the microphysical properties of black carbon. Please cite relevant literature to support this statement.**

In terms of regional and global climate, the observed rapid increase by a factor ~3 in the estimated MAC values needs to be further studied. The reasons for this observed increase require better understanding to be able to quantify by how much the enhanced absorption due to coating is underestimated. If common patterns are found amongst other regional sites, this absorption magnification would need to be considered in climate modeling. In terms of air quality, the expected reduced combustion efficiency could have a role in the observed concentrations of absorbing aerosols. Since high-altitude extreme conditions of hypoxia (leading to increased ventilation) increase the risks associated to air pollution exposure, understanding better the combustion processes and their main sources could help improve and or adjust the locally existing air quality policies.

Finally, absorption measurements from filter-based instruments are extremely susceptible to instrumental artifacts. For this, I would recommend measuring absorption using an independent method to validate the measurements in CHC-GAW to better understand the rapid increase of MAC coefficients, as well as ideally the onsite comparison of the absorption measured after denuding the absorbing particles. As suggested by Virkkula et al. (2021), absorption measurements are incomplete without a continuous monitoring of the size distribution of BC and their coating. I consider that based on the experience gained in the present study, prolonged measurements of rBC on the three sites would largely contribute to untangling the unanswered questions.

The extended discussion of the uniqueness of the LP-EA and CHC-GAW sampling sites, and the implications of our results in regional in global climate previously provided responding to questions 3 and 4 of reviewer #2 were summarized and included in the conclusions section of the manuscript as described below. We thank the reviewer for giving us the opportunity of expanding and improving this section through their questions.

[revised manuscript text omitted]

---

## Author Comment (AC2)

The authors would like to thank the anonymous referee # 1 for taking the time to review the manuscript thoroughly and for appreciating the significance of our study. We thank them for their very relevant comments that allowed us to improve the quality of the manuscript as well as for identifying remaining spelling, numbering and redaction mistakes.

Below you will find the list of the referees' observations (bold), followed right after by the author's responses (normal font) and the respective changes made to the manuscript (italic), highlighting the sections that were modified.

**General comments:**

**Please indicate what "time zone" is used in the manuscript. This is critical especially for interpretation of daily variations. Are the measurements done in UTC or local time? If time is local, how much that deviates from UTC? Was the time shift considered for those data taken from other sources (e.g. EBAS or meteorological observations)?**

We thank the reviewer for pointing out the absence of the specification of the time zone we were referring to when explaining the temporal variations of BC. Indeed, this information is important for interpreting the observations and their link to the local activities. During the analysis the time references of all the datasets handled were carefully considered. To provide more clarity, the following sentence was added in the text.

*"In terms of diurnal variation, a clear bimodal pattern is observed throughout the seasons, which is characteristic of sites influenced by vehicular emissions (Fig. 3 (a)). This pattern has already been described by Wiedensohler et al. (2018) in a shorter temporal scale at EA station (during the transition period to the wet season in 2012). A less pronounced bimodal behavior was observed by the same study in the road site station in La Paz installed during the 2012 short campaign. Henceforth, all the references to time in the present study will be in local time (UTC-4)."*

**Section "2.3 sampling methods" is missing description on sample drying and RH conditions. It is also missing information on how the particulate mass (PM) concentrations were determined, were the filters weighted, which balance and RH were used? How the uncertainties were calculated? This is important since it impacts the eBC mass fraction calculation.**

Unfortunately, no drying mechanism was present in the urban background sites during the campaign. To address more explicitly this limitation in the methodology, the sentence describing the instrumental setup of the aethalometers was modified as follows:

*"Non-dried $PM_{10}$ aerosol particles were sampled at both sites throughout the campaign, except for the period from April 2017 to September 2017 where non-dried whole air was sampled at El Alto station."*

*"In addition, the data recorded between April 2016 and July 2018 by an aethalometer AE31 at CHC-GAW station were downloaded from the EBAS database (http://ebas.nilu.no/) and were included in the analysis. The inlet in CHC-GAW consists of a custom-made, hooded, whole-air inlet that is heated when the relative humidity is >90%%, leading to sampling conditions of RH<40%."*

Moreover, a short description of the methodology used to calculate the PM mass concentrations was added in section 2.3.1., as well as a missing reference on the protocol followed in the measurements of EC from the filter samples. In the added citation, a description of the followed protocol (EUSAAR 2) can be found, which contemplates uncertainties in the range of 2-7%. The paragraph now reads as follows:

*"High-Volume Samplers (MCV CAV-A/mb) were used at both sites to collect aerosol particles on pre-weighed quartz fiber-filters for later analysis. 24-hour filter samples were taken every 3 to 4 days at a flow rate of 30 $m^2h^{-1}$ using $PM_{10}$ heads (MCV PM1025UNE) during the first 15 months of sampling at both sites. For the second year, the head was replaced with a $PM_{2.5}$ inlet (MCV PM1025UNE). Additionally, in La Paz, a second high-volume sampler was added during the second year of the campaign to collect samples of particles with aerodynamic diameters smaller than 10 and 2.5 μm. Sampling always started at 9:00 at both sites. During the campaign, a total of 422 filters were collected between both sites, which were later weighed to quantify the PM mass concentrations through gravimetrical standard procedures (following the EN14907 protocol), conditioning the filters before and after sampling at 20 °C and the RH between 30–35%. The filters were then analyzed for over 180 different chemical species, including EC, OC (through thermal-optical analysis (TOA) using a Sunset instrument and the EUSAAR2 protocol) (Cavalli et al., 2010) and several organic and inorganic source tracers. A more detailed description of the methodology and protocols can be found in Mardoñez et al. (2023)."*

**The last part of the manuscript was slightly challenging to follow due to several missing figures and tables in supplement. Please revise this carefully.**

We sincerely thank the reviewer for identifying the numbering mistakes present in the last section of the manuscript and the unfortunate upload of the wrong version of the Supplementary Information. We added the missing figures and tables in the supplementary information, and we carefully reviewed the references to the images and tables in the text to make sure they correspond to the right numbering of the tables and figures in the manuscript and in the supplementary information.

**Specific comments:**

**Please, check the correct spelling of all the co-authors.**

The spelling of the co-authors' names was reviewed and corrected when necessary.

**L103: "it is also intended" seems like repetition, consider re-phrasing.**

We thank the reviewer for spotting this phrasal repetition. To improve the redaction, the paragraph was slightly modified as follows:

*"This study aims to contribute to document the atmospheric concentration, the variability, and the physical properties of BC in the unique LP-EA conurbation and the global CHC-GAW station. Additionally, it seeks to determine the contribution of local and regional sources of BC in the urban area. To do so, this work makes use of a two-year record of BC and other pollutants measured at two urban background sites and at the mountain CHC-GAW station. The study also provides a spatial description of the BC concentrations and explores the effect of a half-kilometer altitude difference and different topographical characteristics between La Paz and El Alto, thus paving the way for future studies on the potential health effects of air pollution in both cities."*

**L119: What is meant here with temperature amplitude? Max-min?**

We refer as temperature amplitude to the difference between the daily maximum and the daily minimum temperature. To clarify this in the text, a short definition was added next to the first time the term is mentioned.

*"…mean temperatures only increase by one-degree with mean temperature amplitudes (difference between the maximum and the minimum temperature of the day) of 10 and 12 degrees along the day in El Alto and La Paz, respectively"*

**L150: Was the "basic meteorological station" then solely an anemometer? Please clarify & L152: Which "other meteorological variables" were included? These could be listed.**

The paragraph describing the meteorological parameters measured was modified as follows to provide clarity, as suggested by the reviewer:

*"Both urban stations were equipped with anemometers, placed on the rooftop of the buildings where the rest of the instruments were installed, reporting wind speed and wind direction at a 15-min time resolution. For El Alto, pressure, temperature and relative humidity were measured and provided by the Airport's air navigation administration (Administración de Aeropuertos y Servicios Auxiliares a la Navegación Aérea, AASANA) with a 1-hour time resolution, and for La Paz, from the National Meteorology and Hydrology Service (Servicio Nacional de Meteorología e Hidrología, SENAMHI) (SENAMHI, n.d.) at a daily time resolution."*

**L167: replace m2->m3**

We thank the reviewer for spotting this typing error.

**L.265: "the assumption that the properties of the urban aerosol do not change drastically on their transport to CHC-GAW". Which properties of the urban aerosol this refers to? Based on table S2, even during daytime BC aerosol optical properties (MAC) were very different at the sites. How much could this add uncertainty on the defined Cf? Do you have an idea of the SSA at these sites?**

We agree with the reviewer that the term "properties" is rather broad. The real assumption here is that the cross-sensitivity to scattering is somewhat preserved between the urban background and the regional background sites, allowing the extrapolation of the $C_f$ obtained in CHC-GAW to the urban background sites. We consider this assumption to be valid if the parameters that determine $C_f$ are preserved. Among the factors that could most significantly modify $C_f$ would be a change in the filter tape material, a drastic compositional change in the aerosol population, or a change towards very high values of the single scattering albedo (SSA) as described by Yus-Díez et al. (2021). Given that the same model of aethalometer using the same filter tape type was employed at the three sites, the variability due to a difference in the filter media is discarded. Moreover, to constraint the change in the composition of the aerosol population during transport, only the hours in which CHC-GAW is clearly influenced by the urban emissions were considered in the calculation of $C_f$. Finally, although there are no records of the SSA in the urban background sites, they are expected to be lower than the regional background site as for most urban influenced sites, and lower or around the lowest end of threshold interval (0.90-0.95) after which a rapid increase in $C_f$ was observed by Yus-Diez et al. (2021). Thus, having observed that average SSA in CHC-GAW was in average below 0.91 during the studied period, especially during the hours in which the station is influenced by the urban emissions, we consider that our assumption holds.

The text was modified as follows to precise better the assumptions that are being made. Moreover, a figure of the daily variability of SSA was added in the supplementary information.

*"The extrapolation of this calculated $C_f$ factor to the urban background stations is done under the assumption that the cross-sensitivity to scattering does not change drastically between the urban background and the regional background sites. This is supported by the fact that the same model of aethalometer using the same filter type was employed at the three sites, and that the single scattering albedo (SSA) observed in CHC-GAW is in average equal to 0.91 (Figure S11) and even lower during the time interval from 10:00 to 16:00. This specific time frame corresponds to the period in which the CHC-GAW station is typically under the influence of the*

*urban PBL, as previously demonstrated by Andrade et al. (2015)."*

In the supplement the following image was added:

[Figure]

**Figure S11. Diurnal variation of single scattering albedo (SSA) in CHC-GAW, calculated as the ratio of the scattering coefficients (measured by a nephelometer Aurora 3000) to the extinction coefficients (scattering+absorption), during the years 2016-2018. The values included in the figure were constrained to the percentile (1,99) to exclude outliers caused by noise level absorption and scattering coefficients. The shaded area around the solid line represents the 95% confidence interval. The scattering coefficients used to obtain the SSA values displayed in this figure were accessed from EBAS (https://ebas.nilu.no) hosted by NILU. Specifically, the use included data affiliated with the frameworks: ACTRIS, GAW-WDCA.**

**L.279-280: "The babs estimated using the Cf factors selected for each model of AE31 decrease the instrumental difference to roughly 27%." Please, clarify. What are different models of AE31 and what are the instruments for which the difference is 27%?**

We thank the reviewer for pointing out the ambiguity in the mentioned paragraph and the reference mistake to "AE31 models" instead of "aethalometer models". Hoping to provide more clarity, the paragraph was modified in the following way:

*"Systematic higher $b_{abs}$ were observed at EA compared to LP when different models of aethalometers were employed, whereas a negligible difference was observed when using AE31 aethalometers at both urban sites. Moreover, no significant differences in the average concentrations of EC were found among the sites. This indicated that the observed difference was rather associated to differences between models AE33 and AE31. Since an absorption reference instrument was not available to assess the cross-sensitivity to scattering of AE33 at EA, we opted for selecting a suitable literature $C_f$ factor of 2.78 for AE33, for it brought the $b_{abs}$ estimated from both models of aethalometer closer together. The chosen $C_f$ was reported by Bernardoni et al. (2021) at an urban background station in Milan during winter 2018 using the same type of filter tape as AE33 in the present study (T60A20), however, it is 21-40% larger than the values reported by Yus-Díez et al. (2021) for an urban background station in Barcelona using the same tape. Moreover, the $C_f$ values applied in this study are significantly larger than the default values given by the manufacturer of 2.14 and 1.57 for AE31 and AE33, respectively. Therefore, the reported $b_{abs}$ in the present study are smaller than what they would be using the factory default $C_f$ values. The difference between the $b_{abs}$ estimated using the $C_f$ factors selected for each model of aethalometer decrease to roughly 27% when comparing similar periods of measurements in EA."*

**L336. replace WB->BB.**

The typing mistake was corrected

**L341-343: "The pair of wavelengths chosen to apply in this method to apportion the contribution of local vehicular emissions (also known as liquid fuel) from the regional agricultural biomass burning emissions (also known as solid fuel) were 470 and 950 nm." Was there any particular reason for this choice?**

We selected the pair of wavelengths 470 and 950 nm following the recommendations of Zotter et al. (2017). They observed that the $AAE_{TR}$ and $AAE_{BB}$ are not independent of the pair of wavelengths selected to perform the Aethalometer model, and that the best agreement comparing the fossil fuel fraction of eBC obtained from the aethalometer model to the fossil fuel fraction of EC obtained from $^{14}C$ measurements was achieved using the combination 470 and 950 nm.

**L.417 "appear relatively elevated" : compared to what? In previous paragraph the authors write "do not stand out as alarming" which could be interpreted partly contradictory.**

We understand the origin of the apparent contradiction due to the use of the word "relatively" in the observed phrase. We aimed to emphasize that, while the absolute levels of eBC and EC in La Paz-El Alto do not depict the conurbation as a highly polluted metropolitan area, the concentrations appear elevated when compared to those reported from two of the largest high-altitude cities in Latin America. Despite sharing similar emission patterns with Bogota and Mexico City, which have significantly higher populations, the eBC and EC concentrations in La Paz-El Alto are not proportionally lower. We propose the following modification to clarify the statement.

*"As the latter study did not provide information on the temperature and pressure conditions under which the concentrations were measured, these percentages could potentially decrease to approximately 27%-40% if the eBC mass concentrations were reported under ambient conditions (using an annual mean temperature of 15.7 °C and a mean atmospheric pressure of 585 mmHg; Estrada et al., 2009; Hernández-Zenteno et al., 2002). Vehicle emissions have been identified as the main source of EC and eBC in two three metropolitan areas, with vehicle fleets dominated by gasoline fueled vehicles as it is in La Paz and El Alto. However, despite the combined population of the La Paz-El Alto conurbation being more than four times smaller than that reported for Bogota or Mexico City at the time of the studies, the observed concentrations of eBC and EC are not proportionally lower. This phenomenon may be attributed to various factors, including distinctions in population density, vehicle fleet density, combustion efficiencies, the unique topographical features of the cities, notably in La Paz, and pollutant dispersion efficiencies. However, a comprehensive examination of these factors lies outside the scope of the current study.*

**Table 2: Add information on the inlet cut-size (PM10?) after eBC, similar to what is given for EC. Were the 24h-samples of EC for CHC-GAW completely omitted from the analysis? They appear in the methods so please clarify.**

As suggested by the reviewer, we added the size-cut of the inlet placed in front of the Aethalometers, together with a disclaimer indicating the period in which the $PM_{10}$ head was removed at EA, when whole air was sampled.

The 24-h filter samples were initially included because they took part in the calculation of the average EC concentrations in CHC-GAW reported by Moreno et al., 2024. This value was initially included in Table 2 for the annual average concentrations. However, since the number of 24-h filter samples was reduced (5 in total), neither a seasonal description of EC nor representative MAC calculations were feasible for those samples. For this purpose, the 24-h filters were excluded from the rest of the analysis. We understand this could add confusion, for which we propose to remove the term "24-h" in Table 1 of the Methodology, and to modify the average EC concentrations from 0.08 $\mu gm^{-3}$ (which are

the average EC concentrations including the 24-h samples) to 0.07 µgm$^{-3}$ (which are the average EC concentrations including only nighttime samples, from 23:00-08:00). Following this modification, the paragraph discussing this value would exclude the reference to the concentrations reported by Moreno et al., 2024.

*"Table 2 displays the mean STP concentrations of eBC and EC measured in La Paz (LP), El Alto (EA) and Chacaltaya mountain station (CHC-GAW). EC concentrations were similar at both urban sites (EC$_{PM10-LP}$: 2.1±1.2 µg m$^{-3}$, EC$_{PM10-EA}$: 2.4±1.1 µg m$^{-3}$; EC$_{PM2.5-LP}$: 1.5±0.9 µgm$^{-3}$, EC$_{PM2.5-EA}$: 1.6±0.8 µgm$^{-3}$), whereas nocturnal EC concentrations in CHC-GAW were significantly lower (0.07±0.07 µg m$^{-3}$)".*

**L457. replace: suset->sunset**

We thank the reviewer for the remark. It is now corrected.

**L520. "This change in the AAE during this period of the day takes place when eBC mass concentrations reach their minimum, thus, no significant impact in absorption was observed at 880 nm." Message from this sentence is not clear. It raises a question if we are expecting dust to be visible at 880nm?**

The message we were trying to communicate with this sentence is that, given that the study sites are very close to the Bolivian Altiplano, and that previous works have demonstrated dust's large contribution to total PM$_{10}$ mass concentrations (Mardoñez et al., 2023), a priori, dust cannot be ruled out as a potential contributor to absorption. The first analysis made to elucidate its possible impact on absorption measurements was to study the AAE. The observed averaged AAE were rather low, showing that the dominant source of BC in the area is liquid fuel combustion, i.e. traffic. However, analyzing the daily variability of AAE showed that dust's presence can be noted at least as a slight change in the wavelength dependency of absorption happening at midday during the dry season in EA. This observed phenomenon happens while the absorption coefficients reach their daily minimum, for which the impact of dust in total absorption is negligible. This is further confirmed by the source apportionment of the absorption coefficients described in section 2.5.2. using a multilinear regression model, which showed that the significance of the contribution of dust to absorption was low (p-value >0.05) for all wavelengths except 370 nm. Based on this evidence we can conclude that dust do not significantly impact the absorption measurements at 880 nm. We propose the following modification to the text to clarify the message:

*"This change in the wind direction seems to allow the incursion of particles with different optical properties coming from the altiplano, e.g. dust. Given that the change in the AAE during this period of the day takes place when eBC mass concentrations reach their minimum, no significant impact in absorption was observed at 880 nm. This will be further confirmed in section 3.3.2. However, this also reveals that additional sources to traffic and biomass burning could potentially have an impact on absorption and emphasizes the limitation of the aethalometer model to address them."*

**L541. "it overestimated the mixing state of the urban aerosol" here one must remember also the uncertainties in Cf.**

We agree with the reviewer that C$_f$ is not a constant value, and a variation of it could result in a variation of the calculated MAC values. However, the ratio of MAC$_{AE,880}$/MAC$_{EC}$ (~1.3-1.7) is much greater than what a variation of C$_f$ within its range of uncertainty could explain. Since we do not provide a magnitude for this comparison in the text, but just a general tendency, we consider that the sentence remains correct.

**L567. "Nevertheless, the spectral dependence of MACEC and MACrBC values is different." Could this be clarified and opened a bit more?**

The difference between the calculated MAC$_{EC}$ and MAC$_{rBC}$, especially in EA, slightly changes across the wavelengths. However, these variations are found well within the level of uncertainty associated to the calculated magnitudes. The aim of the sentence was to make the reader aware that these differences could be a mere result of the uncertainties in the calculations but could also be impacted by the difference in the period lengths in which the measurements of EC and rBC took place. We recognize that the phrase could bring more confusion than clarity, for which we propose the following modification:

*"From Table 3 it can be observed that despite the EC and rBC concentrations having been measured during different periods of time, with different measuring techniques and different time resolutions, average MAC$_{EC}$ and MAC$_{rBC}$ remain consistent in EA and comparable in CHC. Nevertheless, the small differences between the calculated MAC values for the two techniques measuring mass concentrations (that are found within the levels of uncertainty) could also be influenced by the difference in the sampling period length. Therefore, caution should be taken not to over-interpret them."*

**L591. Fig S6 is missing.**

**L618 Fig S7 is missing.**

**L662-663: Fig S8 and Table S3 are missing**

**L672 S9 missing.**

**L701 Table S4 missing.**

All the missing figures and tables mentioned above were updated in the Supplement.

**Supplement Table S2: replace ii->i**

We thank the reviewer for the remark. It is now corrected.

Moreno, C. I., Krejci, R., Jaffrezo, J., Uzu, G., Alastuey, A., Mardóñez, V., Koenig, A. M., Aliaga, D., Mohr, C., Velarde, F., Blacutt, L., Forno, R., Whiteman, D. N., Wiedensohler, A., Ginot, P., and Laj, P.: Tropical tropospheric aerosol sources and chemical composition observed at high-altitude in the Bolivian Andes, 2837–2860, https://doi.org/https://doi.org/10.5194/acp-24-2837-2024, 2024.

Yus-Díez, J., Bernardoni, V., Močnik, G., Alastuey, A., Ciniglia, D., Ivančič, M., Querol, X., Perez, N., Reche, C., Rigler, M., Vecchi, R., Valentini, S., and Pandolfi, M.: Determination of the multiple-scattering correction factor and its cross-sensitivity to scattering and wavelength dependence for different AE33 Aethalometer filter tapes: a multi-instrumental approach, Atmos. Meas. Tech., 14, 6335–6355, https://doi.org/10.5194/amt-14-6335-2021, 2021.

Zotter, P., Herich, H., Gysel, M., El-Haddad, I., Zhang, Y., Mocnik, G., Hüglin, C., Baltensperger, U., Szidat, S., and Prévôt, A. S. H.: Evaluation of the absorption Ångström exponents for traffic and wood burning in the Aethalometer-based source apportionment using radiocarbon measurements of ambient aerosol, Atmos. Chem. Phys., 17, 4229–4249, https://doi.org/10.5194/acp-17-4229-2017, 2017.